# Algorithmic Recourse for Long-Term Improvement

**Kentaro Kanamori** [1]  **Ken Kobayashi** [2]  **Satoshi Hara** [3]  **Takuya Takagi** [1]

## Abstract

Algorithmic recourse aims to provide a recourse action for altering an unfavorable prediction given by a model into a favorable one (e.g., loan approval). In practice, it is also desirable to ensure that an action makes the real-world outcome better (e.g., loan repayment). We call this requirement *improvement*. Unfortunately, existing methods cannot ensure improvement unless we know the true oracle. To address this issue, we propose a framework for suggesting improvement-oriented actions from a long-term perspective. Specifically, we introduce a new online learning task of assigning actions to a given sequence of instances. We assume that we can observe delayed feedback on whether the past suggested action achieved improvement. Using the feedback, we estimate an action that can achieve improvement for each instance. To solve this task, we propose two approaches based on contextual linear bandit and contextual Bayesian optimization. Experimental results demonstrated that our approaches could assign improvement-oriented actions to more instances than the existing methods.

## 1. Introduction

Machine learning models have been applied to real-world critical decision-making tasks like loan approvals, where predictions significantly impact human users (Rudin, 2019). Thus, decision-makers are expected to explain how users can alter undesired predictions (Miller, 2019; Wachter et al., 2018). *Algorithmic Recourse (AR)* aims to provide such information (Ustun et al., 2019). For a classifier $h\colon \mathcal{X} \to \mathcal{Y}$, a desired class $y^* \in \mathcal{Y}$, and an instance $\boldsymbol{x} \in \mathcal{X}$, AR provides a perturbation vector $\boldsymbol{a}$ that leads the instance $\boldsymbol{x}$ to the desired class $y^*$, i.e., $h(\boldsymbol{x}+\boldsymbol{a}) = y^*$, with a minimum effort measured by some cost function $c$. The user corresponding

---

[1]Fujitsu Limited, Japan [2]Institute of Science Tokyo, Japan [3]The University of Electro-Communications, Japan. Correspondence to: Kentaro Kanamori <k.kanamori@fujitsu.com>.

*Proceedings of the $42^{nd}$ International Conference on Machine Learning*, Vancouver, Canada. PMLR 267, 2025. Copyright 2025 by the author(s).

to $\boldsymbol{x}$ can regard the perturbation $\boldsymbol{a}$ as a *recourse action* for obtaining the desired decision result $y^*$ from the classifier $h$ (Karimi et al., 2022). For example, let us consider a situation where a bank deploys a model $h$ for predicting whether a loan applicant will repay the loan or default, and a user $\boldsymbol{x}$ gets the loan application rejected by $h$. To help the user $\boldsymbol{x}$ get the loan approved, AR suggests an action $\boldsymbol{a}$ that changes the prediction result of $h$ from "default" to "repayment." We call such an action $\boldsymbol{a}$ as *valid with respect to $h$*, or *$h$-valid* for short, across the paper.

While conventional AR methods focus on providing actions that change the prediction result by a model $h$ (i.e., $h$-validity), they often overlook whether the user's real-world outcome actually improves. This gap is critical in real-world applications. For example, in the loan approval scenario, even if the prediction of $h$ changes from "default" to "repayment" by executing the suggested action $\boldsymbol{a}$, the user $\boldsymbol{x}$ may fail to repay the loan if the suggested action $\boldsymbol{a}$ does not actually improve the user's underlying payment capability. Such a gap between the prediction result and real-world outcome can result in degrading the quality and reliability of decision-making (Hardt et al., 2016; Rosenfeld et al., 2020; Estornell et al., 2023; Friedbaum et al., 2024).

To fill this gap, (König et al., 2023) introduced *improvement* as an additional requirement for AR, which focuses on providing actions that improve the user's real-world outcome. Let $h^*\colon \mathcal{X} \to \mathcal{Y}$ be the unknown oracle that returns the real-world outcome corresponding to a given input. (König et al., 2023) claims that actions should be valid with respect to the oracle $h^*$, which we refer to as *$h^*$-validity* to distinguish it from the validity for the classifier $h$. While the classifier $h$ is commonly trained to be a proxy for the oracle $h^*$, it is not completely equivalent to $h^*$ in practice due to various factors such as data limitations, model complexity, and evolving real-world conditions. Thus, since completely filling the gap between $h$ and $h^*$ is unrealistic, $h$-validity does not always imply $h^*$-validity. It indicates that ensuring $h^*$-validity is fundamentally difficult for the conventional formulation of AR that relies solely on $h$.

To address this issue, this paper aims to introduce a new framework of AR for ensuring improvement from a long-term perspective. While the oracle $h^*$ is unknown, it is natural to assume that we can observe the outcome $h^*(\boldsymbol{x}+\boldsymbol{a})$

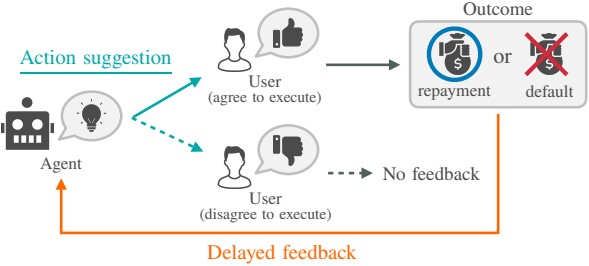

*Figure 1.* Outline of our proposed framework. For a given instance $x$ corresponding to a user, our agent suggests an action $a$ that is estimated as $h^*$-valid for $x$. Then, the user determines whether to execute $a$ based on its cost. If the user executes $a$, the agent observes the delayed feedback on its outcome $h^*(x + a)$. Using the observed feedback, the agent estimates a $h^*$-valid action $a'$ for the subsequent instance $x'$. The agent repeats this procedure.

after a user $x$ executes a suggested action $a$. In the loan approval example, a bank cannot know in advance whether a user $x$ will successfully repay the loan. However, once the loan is approved after executing the suggested action $a$, the bank can monitor whether the user $x$ truly manages to repay it. Thus, we consider a situation in which instances arrive one by one, and for each instance $x$, we suggest an action $a$ and later observe its outcome $h^*(x + a)$, as shown in Figure 1. Our goal is to suggest $h^*$-valid actions with low costs for as many instances as possible by exploring potentially improvement-oriented actions and exploiting the sequentially observed outcomes, which we refer to as *long-term improvement*. To this end, we aim to train an agent that can suggest $h^*$-valid actions for each instance with high probability as the agent obtains more feedback.

### 1.1. Our Contributions

In this paper, we propose *Algorithmic Recourse for Long-Term IMprovement (ARLIM)*, a new framework of AR for ensuring improvement. Our contributions are summarized as follows:

- We introduce a new online learning task where an agent aims to provide $h^*$-valid actions $a_1, \ldots, a_T$ to a given sequence of instances $x_1, \ldots x_T$. In each round $t$, the agent can observe delayed feedback on whether the suggested action $a_s$ was $h^*$-valid for the instance $x_s$ of the past round $s < t$ if $x_s$ has executed $a_s$. Using the feedback, the agent estimates an action $a_t$ that is $h^*$-valid for the current instance $x_t$ with a low cost. The goal of the agent is to assign $h^*$-valid actions with reasonable costs to as many instances as possible.

- For our formulated task, we propose an efficient approach based on the *contextual linear bandit (CLB)*. We show that our task can be reduced to a CLB problem under delayed feedback (Vernade et al., 2020a) if we

know the costs of candidate actions $a$ for each instance $x_t$. Then, we apply the existing efficient algorithm based on LinUCB (Li et al., 2010) to solve our task. We also show that the average improvement achieved by our algorithm is guaranteed to increase with high probability as the number of rounds $t$ progresses.

- We also propose a heuristic approach based on the *contextual Bayesian optimization (CBO)* that can be applicable even if we do not know the cost function $c$. We show that our task can be regarded as a CBO problem under delayed feedback (Verma et al., 2022). To alleviate the scalability issue of the existing algorithm based on the Gaussian process, we propose a scalable algorithm by leveraging a surrogate model based on the extremely randomized trees (Kim & Choi, 2022).

- Our experimental results on real datasets demonstrated that our approaches could achieve improvement for more instances than the existing AR methods. Furthermore, our CLB-based approach could provide actions with comparable costs to the existing methods in the situation where we know the cost function $c$. We also confirmed that when the cost function $c$ includes uncertainty, the performance of our CBO-based approach is close to or better than our CLB-based approach.

### 1.2. Related Work

*Algorithmic Recourse (AR)* (Ustun et al., 2019), also referred to as *Counterfactual Explanation* (Wachter et al., 2018), has attracted increasing attention in recent years. While the previous studies have proposed several extended formulations of AR (Kanamori et al., 2020; 2021; 2022; 2024), almost all of the existing AR methods focus on providing actions that achieve only $h$-validity (Karimi et al., 2022; Verma et al., 2020; Guidotti, 2022). To the best of our knowledge, (König et al., 2023) and (Friedbaum et al., 2024) are the two exceptions. (König et al., 2023) first introduced the concept of improvement, or $h^*$-validity. To evaluate the $h^*$-validity, their proposed method uses a causal graph between features and the target class. However, their method can be applicable only to situations where we are given a true causal graph of the underlying data (Karimi et al., 2020). (Friedbaum et al., 2024) trains a verifier that estimates the $h^*$-validity of each action. As we demonstrate in the experiments, the quality of the verifier hinders the accuracy of this approach.

There exist a few papers that study the *online setting* of AR. For example, (Creager & Zemel, 2023) proposed the setting of online AR where the model $h$ is updated dynamically. However, they focus on updating the model and do not consider long-term improvement. Another example is the work by (Cao et al., 2025) that proposed AR for CLB. In contrast, we study AR for a model $h$ trained by a supervised learning method and use CLB to suggest $h^*$-valid actions.

## 2. Algorithmic Recourse

For a positive integer $n \in \mathbb{N}$, we write $[n] \coloneqq \{1, \ldots, n\}$. As with the previous studies, we consider a binary classification task between undesired and desired classes. Let $\mathcal{X} \subseteq \mathbb{R}^D$ and $\mathcal{Y} = \{0, 1\}$ be input and output domains, respectively. We call a vector $\boldsymbol{x} = (x_1, \ldots, x_D) \in \mathcal{X}$ an *instance*, and a function $h \colon \mathcal{X} \to \mathcal{Y}$ a *classifier*. Without loss of generality, we assume that $h(\boldsymbol{x}) = 1$ is a desirable result. For example, in a loan approval task, $h(\boldsymbol{x}) = 1$ means that a classifier $h$ predicts a user $\boldsymbol{x}$ has the capability to repay the loan.

For an instance $\boldsymbol{x} \in \mathcal{X}$, we define an *action* as a perturbation vector $\boldsymbol{a} \in \mathbb{R}^D$ such that $\boldsymbol{x} + \boldsymbol{a} \in \mathcal{X}$. Let $\mathcal{A}(\boldsymbol{x})$ be a set of feasible actions for $\boldsymbol{x}$ such that $\mathcal{A}(\boldsymbol{x}) \subseteq \{\boldsymbol{a} \in \mathbb{R}^D \mid \boldsymbol{x} + \boldsymbol{a} \in \mathcal{X}\}$. For a classifier $h$, an action $\boldsymbol{a}$ is *valid with respect to $h$*, or *$h$-valid* for an instance $\boldsymbol{x}$ if $h(\boldsymbol{x} + \boldsymbol{a}) = 1$. For $\boldsymbol{x} \in \mathcal{X}$ and $\boldsymbol{a} \in \mathcal{A}(\boldsymbol{x})$, a *cost function* $c \colon \mathcal{A}(\boldsymbol{x}) \to \mathbb{R}_{\geq 0}$ measures the required effort for $\boldsymbol{x}$ to execute $\boldsymbol{a}$.

Given a classifier $h \colon \mathcal{X} \to \mathcal{Y}$ and instance $\boldsymbol{x} \in \mathcal{X}$, *Algorithmic Recourse (AR)* aims to find a feasible action $\boldsymbol{a}$ that is $h$-valid and minimizes its cost $c(\boldsymbol{a} \mid \boldsymbol{x})$ for $\boldsymbol{x}$. It can be formulated as follows (Karimi et al., 2022):

$$\min_{\boldsymbol{a} \in \mathcal{A}(\boldsymbol{x})} c(\boldsymbol{a} \mid \boldsymbol{x}) \quad \text{s.t.} \quad h(\boldsymbol{x} + \boldsymbol{a}) = 1. \qquad (1)$$

Throughout this paper, we assume that a set of feasible actions can be expressed as $\mathcal{A}(\boldsymbol{x}) = [l_1, u_1] \times \cdots \times [l_D, u_D]$ with lower and upper bounds $l_d, u_d \in \mathbb{R}$ for $d \in [D]$. For example, we can express an immutable feature (e.g., gender) by setting $l_d = u_d = 0$, and a feature that is allowed to be only increased (e.g., education level) by $l_d = 0$ and $u_d > 0$, respectively. We also assume that a classifier $h$ is fixed.

## 3. Problem Formulation

The goal of this paper is to suggest an action $\boldsymbol{a}$ that achieves *improvement* (König et al., 2023) for a given instance $\boldsymbol{x}$. Let $h^* \colon \mathcal{X} \to \mathcal{Y}$ be the unknown oracle that maps an input to its real-world outcome. Then, ensuring improvement is equivalent to satisfying $h^*(\boldsymbol{x} + \boldsymbol{a}) = 1$, i.e., *$h^*$-validity*. In practice, however, we can not evaluate the $h^*$-validity for arbitrary $\boldsymbol{x}$ and $\boldsymbol{a}$ in advance. Thus, it is fundamentally difficult for the existing formulation (1) to ensure improvement.

To ensure improvement, we extend the existing problem setting of AR from the long-term perspective. While the oracle $h^*$ is unknown, it is natural to assume that we can observe the outcome $h^*(\boldsymbol{x} + \boldsymbol{a})$ after a user $\boldsymbol{x}$ executes a suggested action $\boldsymbol{a}$, as described in Section 1. Thus, we consider a situation in which instances $\boldsymbol{x}_1, \boldsymbol{x}_2, \ldots, \boldsymbol{x}_T$ arrive one by one. For each instance $\boldsymbol{x}_t$, we suggest an action $\boldsymbol{a}_t$ and later observe delayed feedback on its outcome $h^*(\boldsymbol{x}_t + \boldsymbol{a}_t)$, as shown in Figure 1. By exploring potentially improvement-oriented actions and exploiting the sequentially observed

outcomes, we aim to suggest actions that are both $h$- and $h^*$-valid with low costs for as many instances as possible, which we refer to as *long-term improvement*. In this section, we formulate this task as an online learning problem.

### 3.1. Reward Model and Stochastic Delay

To formulate our task, we define our criterion for evaluating an action $\boldsymbol{a}_t$ assigned to an instance $\boldsymbol{x}_t$ as a *reward*. In this paper, we assume that we can observe the outcome $h^*(\boldsymbol{x}_t + \boldsymbol{a}_t)$ following the unknown distribution $P(Y \mid X = \boldsymbol{x}_t + \boldsymbol{a}_t)$ if the suggested action $\boldsymbol{a}_t$ is executed by $\boldsymbol{x}_t$. However, the user $\boldsymbol{x}_t$ may not execute $\boldsymbol{a}_t$ if its cost $c(\boldsymbol{a}_t \mid \boldsymbol{x}_t)$ is higher than expected (Wu et al., 2024), which makes its outcome $h^*(\boldsymbol{x}_t + \boldsymbol{a}_t)$ unobservable. To encourage the user $\boldsymbol{x}_t$ to execute the suggested action $\boldsymbol{a}_t$, its cost $c(\boldsymbol{a}_t \mid \boldsymbol{x}_t)$ should be as low as possible. Therefore, our reward model needs to take into account the cost $c$, as well as $h^*$-validity.

Based on the above motivation, we define our reward model. Let $\mathcal{B}(p)$ be the Bernoulli distribution with $p \in [0, 1]$. Then, we define a reward $R_t \in \{0, 1\}$ of assigning an action $\boldsymbol{a}_t$ to an instance $\boldsymbol{x}_t$ of a round $t$ as follows:

$$R_t \sim \mathcal{B}(E(\boldsymbol{a}_t \mid \boldsymbol{x}_t) \cdot I(\boldsymbol{a}_t \mid \boldsymbol{x}_t)), \qquad (2)$$

where $E(\boldsymbol{a} \mid \boldsymbol{x})$ and $I(\boldsymbol{a} \mid \boldsymbol{x})$ are the probabilities that an instance $\boldsymbol{x}$ executes $\boldsymbol{a}$ and that an action $\boldsymbol{a}$ achieves improvement (i.e., $h^*$-validity) for $\boldsymbol{x}$, respectively. Following the existing work (Fokkema et al., 2024), we define the former as $E(\boldsymbol{a} \mid \boldsymbol{x}) \coloneqq \exp(-\nu \cdot c(\boldsymbol{a} \mid \boldsymbol{x}))$ with a parameter $\nu > 0$. By definition, the executing probability $E(\boldsymbol{a} \mid \boldsymbol{x})$ of an action $\boldsymbol{a}$ decreases exponentially in its cost $c(\boldsymbol{a} \mid \boldsymbol{x})$. We define the latter as $I(\boldsymbol{a} \mid \boldsymbol{x}) \coloneqq P(Y = 1 \mid X = \boldsymbol{x} + \boldsymbol{a})$, which is unknown for us. Note that we can relax the definition of the improvement probability $I(\boldsymbol{a} \mid \boldsymbol{x})$ to the deterministic one by setting $I(\boldsymbol{a} \mid \boldsymbol{x}) = h^*(\boldsymbol{x} + \boldsymbol{a})$, which is a special case of our definition of $I(\boldsymbol{a} \mid \boldsymbol{x})$. In a nutshell, if an action $\boldsymbol{a}_t$ is $h^*$-valid for an instance $\boldsymbol{x}_t$ and its cost $c(\boldsymbol{a}_t \mid \boldsymbol{x}_t)$ is low, its reward $R_t$ takes 1 with high probability.

In our scenario, it is natural that we can not immediately observe the reward $R_t$ after assigning $\boldsymbol{a}_t$ to $\boldsymbol{x}_t$. This is because there are delays until the instance $\boldsymbol{x}_t$ executes $\boldsymbol{a}_t$ and its outcome $h^*(\boldsymbol{x}_t + \boldsymbol{a}_t)$ becomes observable. To model such delays, we consider the *stochastic delayed feedback setting* (Vernade et al., 2017) where we observe the reward $R_t$ of a round $t$ at the future round $t + D_t$, where $D_t \in \mathbb{N}$ is a delay variable drawn from an unknown distribution $\mathcal{D}$.

### 3.2. Overall Problem Setup

Using our reward model, we formulate our task as an online learning problem between an agent and an environment. In each round $t \in [T]$, the agent receives an instance $\boldsymbol{x}_t$ and selects an action $\boldsymbol{a}_t$ among feasible and $h$-valid actions (i.e., $\boldsymbol{a}_t \in \mathcal{A}(\boldsymbol{x}_t)$ and $h(\boldsymbol{x}_t + \boldsymbol{a}_t) = 1$). Then, the environment

samples a reward $R_t$ following our reward model defined in (2). The goal of the agent is to maximize the mean expected reward $\mathcal{R}(T) := \frac{1}{T} \sum_{t=1}^{T} \mathbb{E}[R_t]$; that is, to assign actions $\boldsymbol{a}_t$ that achieve $h^*$-validity with low costs to as many instances $\boldsymbol{x}_t$ as possible. Our task, named *Algorithmic Recourse for Long-Term IMprovement (ARLIM)*, is formulated as follows.

**Problem 3.1** (Algorithmic Recourse for Long-Term IMprovement, ARLIM)**.** We assume a classifier $h: \mathcal{X} \to \mathcal{Y}$. For each round $t \in \{1, 2, \ldots, T\}$, the following procedure is repeated between an agent and environment:

1. The agent receives an instance $\boldsymbol{x}_t \in \mathcal{X}$ with $h(\boldsymbol{x}) = 0$ from the environment, and constructs a set of candidate actions $\mathcal{A}_t$ that are feasible and $h$-valid for $\boldsymbol{x}_t$.

2. The agent selects an action $\boldsymbol{a}_t \in \mathcal{A}_t$ based on past observations and sends it to the environment.

3. The environment samples a reward $R_t$ and delay $D_t$ from $\mathcal{B}(E(\boldsymbol{a}_t \mid \boldsymbol{x}_t) \cdot I(\boldsymbol{a}_t \mid \boldsymbol{x}_t))$ and $\mathcal{D}$, respectively.

4. The agent observes a set of the past rewards $\{R_s \mid s + D_s = t\}_{s=1}^{t-1}$ that become observable at round $t$.

The goal of the agent is to maximize the mean expected reward $\mathcal{R}(T) = \frac{1}{T} \sum_{t=1}^{T} \mathbb{E}[R_t]$ over the total rounds $T$.

By solving Problem 3.1, we are expected to suggest recourse actions that achieve improvement with reasonable costs for as many instances as possible. In Sections 4 and 5, we propose two practical approaches for solving Problem 3.1.

# 4. Contextual Linear Bandit Approach

In this section, we propose an algorithm to solve Problem 3.1 based on the *contextual linear bandit (CLB)* approach. We show that Problem 3.1 can be reduced to the *contextual linear bandit problem under stochastic delayed feedback* (Vernade et al., 2020a) if we can compute the executing probability $E$ of each candidate action $\boldsymbol{a} \in \mathcal{A}_t$. Then, we apply the existing efficient algorithm based on LinUCB to solve Problem 3.1 and provide its theoretical analyses. All the proofs of the statements are presented in Appendix A.

## 4.1. Basic Idea

To show that Problem 3.1 can be reduced to a CLB problem, we make some assumptions on the candidate actions $\mathcal{A}_t$ and the executing probability $E$. First, to construct $\mathcal{A}_t$ in each round $t$, we follow the existing AR methods based on the prototypes (Van Looveren & Klaise, 2021) or nearest unlike neighbors (Brughmans et al., 2024). We assume that the agent is given a fixed set of $K$ instances $\tilde{\mathcal{X}} = \{\tilde{\boldsymbol{x}}_1, \ldots, \tilde{\boldsymbol{x}}_K\}$ with $h(\tilde{\boldsymbol{x}}_k) = 1$ for any $k \in [K]$, which we call *recourse instances*. For $k \in [K]$, we denote by $\boldsymbol{a}_k^{(t)} := \tilde{\boldsymbol{x}}_k - \boldsymbol{x}_t$.

Using $\tilde{\mathcal{X}}$, we construct $\mathcal{A}_t$ as follows:

$$\mathcal{A}_t = \mathcal{A}(\boldsymbol{x}_t) \cap \{\boldsymbol{a}_1^{(t)} \ldots \boldsymbol{a}_K^{(t)}\}. \tag{3}$$

Note that any action $\boldsymbol{a} \in \mathcal{A}_t$ is feasible and $h$-valid for $\boldsymbol{x}_t$ by definition. We also assume that the agent knows the cost function $c$ and parameter $\nu$. It means that the agent can compute $E(\boldsymbol{a} \mid \boldsymbol{x}_t)$ for any $\boldsymbol{a} \in \mathcal{A}_t$. Note that any existing cost function can be used as $c$, such as $\ell_1$-norm (Wachter et al., 2018) or max percentile shift (Ustun et al., 2019).

Under the above assumptions, we show that Problem 3.1 can be reduced to a variant of the CLB problem.

**Proposition 4.1.** *We assume that $\mathcal{A}_t$ of each $t \in [T]$ is constructed as (3) and the cost function $c$ and parameter $\nu$ are known. Then, Problem 3.1 is reduced to a CLB problem under stochastic delayed feedback (Vernade et al., 2020a).*

*Proof (sketch).* We denote $\mathcal{A}_t = \{\boldsymbol{a}_1, \ldots, \boldsymbol{a}_L\}$. By definition, there exists a mapping $\pi_t: [L] \to [K]$ such that $\boldsymbol{x}_t + \boldsymbol{a}_l = \tilde{\boldsymbol{x}}_{\pi_t(l)}$ for any $l \in [L]$. Let $\phi_t(\boldsymbol{a}_l) := E(\boldsymbol{a}_l \mid \boldsymbol{x}_t) \cdot \boldsymbol{e}_{\pi_t(l)}$, where $\boldsymbol{e}_k \in \{0, 1\}^K$ is the binary vector where its $k$-th element is 1 and the others are 0. By definition, we have $I(\boldsymbol{a}_l \mid \boldsymbol{x}_t) = P(Y = 1 \mid X = \tilde{\boldsymbol{x}}_{\pi_t(l)})$. We denote $\boldsymbol{\theta} := (P(Y = 1 \mid X = \tilde{\boldsymbol{x}}_1), \ldots, P(Y = 1 \mid X = \tilde{\boldsymbol{x}}_K))$ and $A_t := \phi_t(\boldsymbol{a}_t)$, respectively. Then, we have $A_t^\top \boldsymbol{\theta} = E(\boldsymbol{a}_t \mid \boldsymbol{x}_t) \cdot I(\boldsymbol{a}_t \mid \boldsymbol{x}_t)$. Therefore, our reward $R_t$ of the round $t$ can be expressed as $R_t \sim \mathcal{B}(A_t^\top \boldsymbol{\theta})$. Furthermore, $\phi_t(\boldsymbol{a})^\top \boldsymbol{\theta} \in [0, 1]$ and $\|\phi_t(\boldsymbol{a})\|_2 \leq 1$ hold for any $\boldsymbol{a} \in \mathcal{A}_t$. The above facts and the definition imply that Problem 3.1 is reduced to the CLB problem under stochastic delayed feedback defined in (Vernade et al., 2020a). $\square$

Proposition 4.1 shows that Problem 3.1 can be reduced to the CLB problem under stochastic delayed feedback if how to construct the candidate actions $\mathcal{A}_t$ is limited to (3) and the executing probability $E$ is known for the agent. It implies that we can apply the existing algorithm with a regret guarantee (Vernade et al., 2020a) to solve Problem 3.1.

## 4.2. Algorithm

Algorithm 1 presents our algorithm for Problem 3.1 based on the CLB. As with the proof of Proposition 4.1, we define $\phi_t(\boldsymbol{a}_l) := E(\boldsymbol{a}_l \mid \boldsymbol{x}_t) \cdot \boldsymbol{e}_{\pi_t(l)}$. Algorithm 1 is based on the OTFLinUCB algorithm proposed by (Vernade et al., 2020a), which is an extension of the well-known LinUCB algorithm (Abbasi-Yadkori et al., 2011) to the stochastic delayed feedback setting. In the delayed feedback setting, it is not suitable to wait for the rewards of all the past actions until they are observed. Hence, the OTFLinUCB algorithm uses a sliding window of size $m$ and ignores an unobserved reward $R_s$ whose delay $D_s$ exceeds $m$ by regarding $R_s = 0$. Using such censored rewards, Algorithm 1 estimates the unknown parameter $\boldsymbol{\theta}$ by $\ell_2$-regularized least squares with

**Algorithm 1** LinUCB

**Input:** Set of $K$ recourse instances $\tilde{\mathcal{X}}$, window parameter $m > 0$, regularization parameter $\lambda > 0$, and confidence level $\delta > 0$.

1: $V_1^{-1} \leftarrow \lambda \cdot I$; $\boldsymbol{b}_1 \leftarrow \boldsymbol{0}$;
2: **for** $t = 1, \ldots, T$ **do**
3:     Receive $\boldsymbol{x}_t$ and construct $\mathcal{A}_t$ by (3);
4:     $V_t^{-1} \leftarrow V_{t-1}^{-1} + \frac{V_{t-1}^{-1} A_{t-1} A_{t-1}^\top V_{t-1}^{-1}}{1 + A_{t-1}^\top V_{t-1}^{-1} A_{t-1}}$;
5:     $\hat{\boldsymbol{\theta}}_t \leftarrow V_t^{-1} \boldsymbol{b}_t$;
6:     $\alpha_t \leftarrow \sqrt{\lambda} + \sqrt{2 \cdot \log{(1/\delta)} + K \cdot \log{(1 + t/K\lambda)}}$;
7:     $\kappa_t \leftarrow 2 \cdot \alpha_t + \sum_{s=t-m}^{t-1} A_s^\top V_t^{-1} A_s$;
8:     $\boldsymbol{a}_t \leftarrow \arg\max_{\boldsymbol{a} \in \mathcal{A}_t} \phi_t(\boldsymbol{a})^\top \hat{\boldsymbol{\theta}}_t + \kappa_t \cdot \phi_t(\boldsymbol{a})^\top V_t^{-1} \phi_t(\boldsymbol{a})$;
9:     Suggest $\boldsymbol{a}_t$ and set $A_t \leftarrow \phi_t(\boldsymbol{a}_t)$;
10:    Observe $\{R_s \mid s + D_s = t\}_{s=t-m}^{t-1}$;
11:    $\boldsymbol{b}_{t+1} \leftarrow \boldsymbol{b}_t + \sum_{s=t-m}^{t-1} R_s \cdot \mathbb{I}[s + D_s = t] \cdot A_s$;
12: **end for**

a given regularization parameter $\lambda > 0$. Then, it determines an action $\boldsymbol{a}_t$ with the estimated parameter $\hat{\boldsymbol{\theta}}$ and a modified upper confidence bound that takes into account the bias caused by unobserved rewards.

### 4.3. Theoretical Analysis

Using the regret bound of the OTFLinUCB algorithm (Vernade et al., 2020a), we show a lower bound on the mean expected reward $\mathcal{R}(T)$ that can be achieved by Algorithm 1.

**Proposition 4.2.** *Algorithm 1 satisfies the following inequality with probability $1 - \delta$ for any $\delta > 0$:*

$$\mathcal{R}(T) \geq \mathcal{R}_T^* - \Gamma_T, \tag{4}$$

*where $\mathcal{R}_T^* = \frac{1}{T} \sum_{t=1}^{T} \max_{\boldsymbol{a} \in \mathcal{A}_t} E(\boldsymbol{a} \mid \boldsymbol{x}_t) \cdot I(\boldsymbol{a} \mid \boldsymbol{x}_t)$, $\Gamma_T = \frac{4}{T \cdot \tau_m}(\alpha_T \sqrt{2KT \log \gamma_T} + mK \log \gamma_T)$, $\tau_m = P(D_1 \leq m)$, $\alpha_T = \sqrt{\lambda} + \sqrt{2 \log{(1/\delta)} + K \log \gamma_T}$, and $\gamma_T = 1 + T/K\lambda$.*

Proposition 4.2 implies that the mean expected reward attained by Algorithm 1 converges to its upper bound with high probability as increasing $T$. The first term $\mathcal{R}_T^*$ is the empirical average of the best expected reward, which is the upper bound of $\mathcal{R}(T)$. In addition, we have roughly $\Gamma_T = \mathcal{O}(\log T / \sqrt{T})$ and thus $\Gamma_T$ converges to 0 as increasing $T$. This result indicates that the longer our algorithm is deployed, the higher its probability of suggesting actions that achieve improvement with reasonable costs is.

Note that the delay distribution $\mathcal{D}$ impacts the term $\tau_m$ in our bound of Proposition 4.2. This term increases as the delay $D_1 \sim \mathcal{D}$ for the first instance $\boldsymbol{x}_1$ tends to be smaller than the window parameter $m$ of our algorithm, which makes our bound better. In essence, the more quickly feedback is received, the better our algorithm performs. In addition, if the delay $D_m$ of each round $t$ is some fixed value and we know it in advance, we can set our window parameter $m$ to

that value. This adaptation would likely lead to improved performance compared to the stochastic delay setting.

**Time Complexity.** We can analyze the time complexity of Algorithm 1 in each round $t \in [T]$ as follows. From the equation (3) and assumption on $\mathcal{A}(\boldsymbol{x}_t)$, we can construct $\mathcal{A}_t$ in $\mathcal{O}(K \cdot D)$ time. In addition, by the definition of $\phi_t$, we can compute (i) $V_t^{-1}$ and $\hat{\boldsymbol{\theta}}_t$ in $\mathcal{O}(K^2)$; (ii) $\kappa_t$ and $\boldsymbol{b}_{t+1}$ in $\mathcal{O}(m)$; (iii) $\boldsymbol{a}_t$ in $\mathcal{O}(K)$. In total, the time complexity of Algorithm 1 in each round is $\mathcal{O}(K \cdot D + K^2 + m)$.

## 5. Contextual Bayesian Optimization Approach with BwO Forest

In this section, we propose an algorithm to solve Problem 3.1 based on the *contextual Bayesian optimization (CBO)* approach. In contrast to the CLB approach in Section 4, this approach does not require the executing probability $E$ to be known. To this end, we regard Problem 3.1 as the *contextual Bayesian optimization problem under stochastic delayed feedback* (Verma et al., 2022). Then, we propose a UCB-based heuristic algorithm that leverages extremely randomized trees as a surrogate model (Kim & Choi, 2022).

### 5.1. Basic Idea

Let $g \colon \mathcal{X} \times \mathcal{A} \to [0, 1]$ be an unknown function such that $R_t = g(\boldsymbol{x}_t, \boldsymbol{a}_t) + \varepsilon_t$, where $\mathcal{A} = \mathcal{A}_1 \cup \cdots \cup \mathcal{A}_T$ and $\varepsilon_t \in \mathbb{R}$ is a sub-Gaussian noise. Then, we can regard Problem 3.1 as the CBO problem under stochastic delayed feedback (Verma et al., 2022) with the unknown objective function $g$. It indicates that we are expected to apply the existing algorithm that is based on the Gaussian process (GP) and has theoretical guarantees (Verma et al., 2022) to Problem 3.1. Unfortunately, in our preliminary experiments shown in Appendix B, we confirmed that the existing algorithm based on GP was significantly slower compared to Algorithm 1.

To improve scalability, we employ the *bagging with oversampling (BwO) forest* (Kim & Choi, 2022) as a surrogate model instead of GP. The BwO forest is a variant of tree-based ensemble models that elaborates prediction uncertainty estimation. Tree-based surrogate models are known to have practical merits over GP due to their scalability and ability to naturally deal with categorical features (Hutter et al., 2011). While the BwO forest was originally proposed for sequential model-based optimization, it is also suitable as a surrogate model in the CBO problem since it can estimate the mean and variance of the reward for an instance-action pair $(\boldsymbol{x}, \boldsymbol{a}) \in \mathcal{X} \times \mathcal{A}$.

Each decision tree $f \colon \mathcal{X} \times \mathcal{A} \to [0, 1]$ in the BwO forest is trained by bootstrap sampling with oversampling. In each node of the tree, both a split feature and threshold are selected randomly, as with the extremely randomized

**Algorithm 2** BwOUCB

**Input:** Set of $K$ recourse instances $\tilde{\mathcal{X}}$, window parameter $m > 0$, and number of trees $B$.
1: Initialize a BwO forest $\mathcal{F} = \{f_1, \ldots, f_B\}$ with $B$ trees;
2: $\mathcal{Z} \leftarrow \emptyset$;
3: **for** $t = 1, \ldots, T$ **do**
4:     Receive $\boldsymbol{x}_t$ and construct $\mathcal{A}_t$;
5:     $\kappa_t \leftarrow \sqrt{(1/t) \cdot \log t}$;
6:     $\boldsymbol{a}_t \leftarrow \arg\max_{\boldsymbol{a} \in \mathcal{A}_t} \hat{\mu}(\boldsymbol{x}_t, \boldsymbol{a}; \mathcal{F}) + \kappa_t \cdot \hat{\sigma}(\boldsymbol{x}_t, \boldsymbol{a}; \mathcal{F})$;
7:     $z_t \leftarrow (\boldsymbol{x}_t, \boldsymbol{a}_t, 0)$; $\mathcal{Z} \leftarrow \mathcal{Z} \cup \{z_t\}$;
8:     Suggest $\boldsymbol{a}_t$;
9:     Observe $\{R_s \mid s + D_s = t\}_{s=t-m}^{t-1}$;
10:     **for** $s = t - m, \ldots, t - 1$ **do**
11:         **if** $s + D_s = t$ and $R_s = 1$ **then**
12:             $\mathcal{Z} \leftarrow \mathcal{Z} \setminus \{z_s\}$; $\mathcal{Z} \leftarrow \mathcal{Z} \cup \{(\boldsymbol{x}_s, \boldsymbol{a}_s, R_s)\}$;
13:         **end if**
14:     **end for**
15:     Train $\mathcal{F}$ using $\mathcal{Z}$;
16: **end for**

trees (Geurts et al., 2006). Such a randomized strategy is known to improve scalability without significantly degrading generalization performance. Using an ensemble of $B$ trained trees $\mathcal{F} = \{f_1, \ldots, f_B\}$, the BwO forest estimates the mean and variance of the reward of $(\boldsymbol{x}, \boldsymbol{a})$ as follows:

$$\hat{\mu}(\boldsymbol{x}, \boldsymbol{a}; \mathcal{F}) = \frac{1}{B} \sum_{b=1}^{B} \sum_{l=1}^{L_b} \mu_{b,l} \cdot \pi_{b,l}(\boldsymbol{x}, \boldsymbol{a}),$$

$$\hat{\sigma}^2(\boldsymbol{x}, \boldsymbol{a}; \mathcal{F}) = \frac{1}{B} \sum_{b=1}^{B} \sum_{l=1}^{L_b} \left(\sigma_{b,l}^2 + \mu_{b,l}^2\right) \cdot \pi_{b,l}(\boldsymbol{x}, \boldsymbol{a}) - (\hat{\mu}(\boldsymbol{x}, \boldsymbol{a}; \mathcal{F}))^2,$$

where $L_b$ is the total number of leaves in $f_b$, $\pi_{b,l} \colon \mathcal{X} \times \mathcal{A} \to \{0, 1\}$ is the indicator whether a sample $(\boldsymbol{x}, \boldsymbol{a})$ reaches the leaf $l$ of $f_b$, and $\mu_{b,l}$ and $\sigma_{b,l}^2$ are the mean and variance of the training samples that reach the leaf $l$ of $f_b$, respectively.

### 5.2. Algorithm

Algorithm 2 presents our algorithm for Problem 3.1 based on the CBO with the BoW forest. Algorithm 2 is an extension of the GP-UCB-SDF algorithm proposed by (Verma et al., 2022). We replace its surrogate model based on GP with that based on the BoW forest (Kim & Choi, 2022) so as to handle our problem. The surrogate model $\mathcal{F}$ is trained using the past histories $\mathcal{Z}$, which is a set of tuples consisting of an instance $\boldsymbol{x}_s$, action $\boldsymbol{a}_s$, and reward $R_s$ for the past round $s < t$. As with OTFLinUCB, GP-UCB-SDF uses a sliding window of size $m$ and replaces unobserved rewards whose delays exceed $m$ with $R_s = 0$. In each round $t$, Algorithm 2 determines an action $\boldsymbol{a}_t$ based on the upper confidence bound score estimated by the BwO forest $\mathcal{F}$.

### 5.3. Comparison to Contextual Linear Bandit Approach

Here, we discuss the pros and cons of Algorithm 2 compared to Algorithm 1. One main advantage of Algorithm 2 is that

it does not require the executing probability $E$ to be known, which is often difficult to estimate due to user preferences and uncertainties in the cost function $c$ (Laugel et al., 2023; Toni et al., 2024). The BwO forest in Algorithm 2 estimates both $E$ and $I$ from past observations. We also note that Algorithm 2 has no constraints on constructing $\mathcal{A}_t$.

However, Algorithm 2 lacks the theoretical guarantees that Algorithm 1 has, as shown in Proposition 4.2. Additionally, Algorithm 2 requires retraining the BwO forest $\mathcal{F}$ in each round, which is computationally expensive compared to the $\ell_2$-regularized least squares of Algorithm 1. In summary, Algorithm 2 is more practical in situations where estimating $E$ is challenging, such as when the user's demographic features are diverse (Tominaga et al., 2024; Venkatasubramanian & Alfano, 2020; Sullivan & Verreault-Julien, 2022). In contrast, Algorithm 1 is more suitable for tasks with less uncertainty due to its theoretical guarantees and efficiency.

## 6. Experiments

To investigate the efficacy of our framework, we conducted experiments on real datasets. All the code was implemented in Python 3.10 and is available at https://github.com/kelicht/arlim. All the experiments were conducted on macOS Sequoia with Apple M2 Ultra CPU and 128 GB memory. Our experimental evaluation aims to answer the following questions: (i) How are the improvement and cost of the recourse actions suggested by our algorithms compared to the existing baselines? (ii) Can our algorithms become to suggest better actions as the round progresses? (iii) In which situations is the performance of our CBO-based algorithm (Algorithm 2) close to that of our CLB-based algorithm (Algorithm 1)? Due to page limitations, the complete results are shown in Appendix B.

### 6.1. Experimental Settings

**Datasets.** We used three real-world datasets: Credit ($N = 30000, D = 13$) (Yeh & hui Lien, 2009), Diabetes ($N = 769, D = 8$) (Dua & Graff, 2017), and COMPAS ($N = 6167, D = 9$) (Angwin et al., 2016). All the categorical features in each dataset were one-hot encoded, and all the numerical features were normalized to $[0, 1]$. Details of the datasets and our preprocessing are shown in Appendix B.

**Protocol.** Since we do not know the oracle $h^*$ in each real dataset, we can not evaluate the outcome $h^*(\boldsymbol{x}_t + \boldsymbol{a}_t)$ if we allow arbitrary actions $\boldsymbol{x}_t$ for $\boldsymbol{a}_t$. However, if there exists a sample $(\tilde{\boldsymbol{x}}, y)$ in the dataset such that $\tilde{\boldsymbol{x}} = \boldsymbol{x}_t + \boldsymbol{a}_t$, we can regard the label $y$ of $\tilde{\boldsymbol{x}}$ as a proxy for the oracle outcome $h^*(\boldsymbol{x}_t + \boldsymbol{a}_t)$. Based on this idea, we restrict the candidate actions $\boldsymbol{a}_t \in \mathcal{A}_t$ to those that can shift to one of the recourse instances $\tilde{\boldsymbol{x}} \in \tilde{\mathcal{X}}$. More precisely, we randomly collect some instances $\tilde{\boldsymbol{x}}$ from the dataset as $\tilde{\mathcal{X}}$ and construct $\mathcal{A}_t$

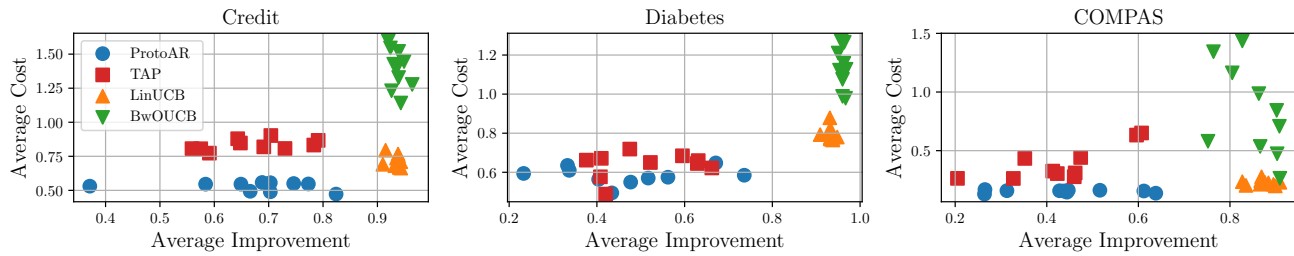

(a) Average Improvement (higher is better) and Average Cost (lower is better)

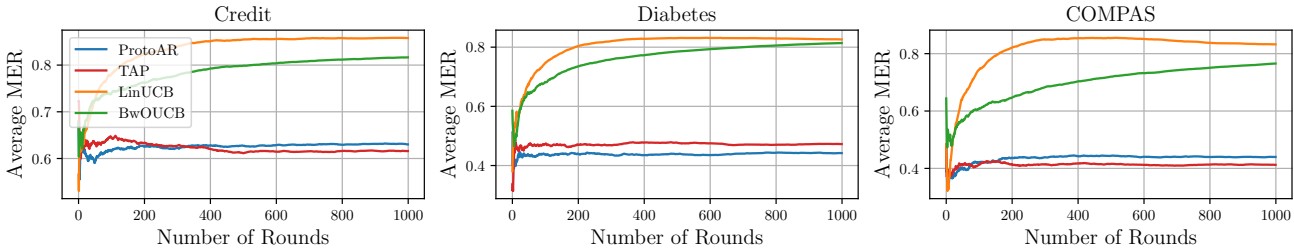

(b) Average Mean Expected Reward (MER) in Each Round (higher is better)

*Figure 2.* Experimental results of baseline comparison under the noiseless cost evaluation situation. Our LinUCB and BwOUCB attained higher improvement than the baselines, and their the mean expected reward increased as the round progressed.

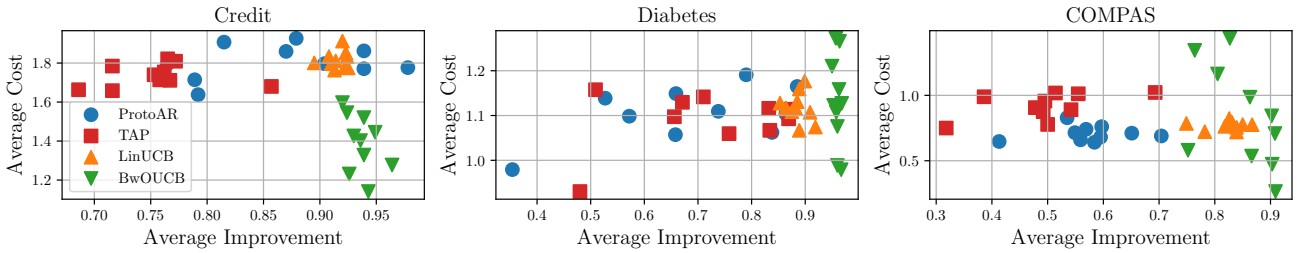

(a) Average Improvement (higher is better) and Average Cost (lower is better)

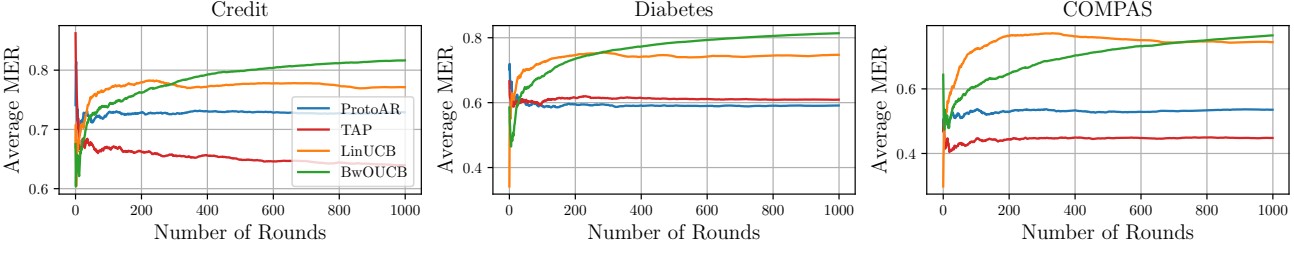

(b) Average Mean Expected Reward (MER) in Each Round (higher is better)

*Figure 3.* Experimental results of baseline comparison under the noisy cost evaluation situation with the noise level $\xi = 0.25$. Compared to the results under the noiseless situation, we observed that the performance of BwOUCB was better than or close to that of LinUCB.

by following (3) with $\tilde{\mathcal{X}}$. By constraining each method to select an action $\boldsymbol{a}_t$ from $\mathcal{A}_t$, we can evaluate $h^*(\boldsymbol{x}_t + \boldsymbol{a}_t)$ by the label $y$ of $\tilde{\boldsymbol{x}}$ such that $\tilde{\boldsymbol{x}} = \boldsymbol{x}_t + \boldsymbol{a}_t$. To simulate the noise of the outcome, we set its improvement probability $I(\boldsymbol{a}_t \mid \boldsymbol{x}_t)$ by adding a random noise $\varepsilon$ to $y$ and scaling it in $[0, 1]$. In summary, we conducted the following procedures:

1. We randomly split the dataset $S = \{(\boldsymbol{x}_n, y_n)\}_{n=1}^N$ into the training set $S_{\mathrm{tr}}$, recourse set $S_{\mathrm{re}}$, and test set $S_{\mathrm{te}}$

with a ratio of $2 : 1 : 1$.

2. We trained a classifier $h$ on the training set $S_{\mathrm{tr}}$. As $h$, we used a random forest (RF) with 100 trees or a two-layer neural network (NN) with 100 neurons.

3. We constructed the recourse instance set $\tilde{\mathcal{X}}$ with maximum size $K = 200$ by randomly collecting the instances $\boldsymbol{x}_n$ in the recourse set $S_{\mathrm{re}}$ such that $h(\boldsymbol{x}_n) = 1$.

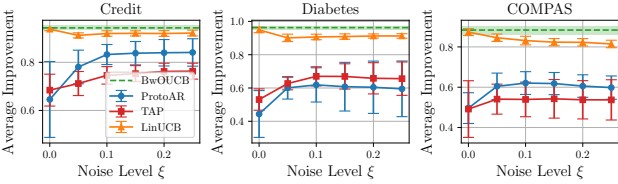

(a) Average Improvement (higher is better)

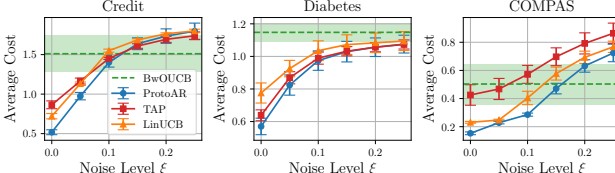

(b) Average Cost (lower is better)

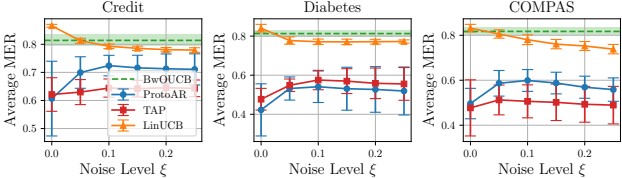

(c) Average MER of the Final Round (higher is better)

*Figure 4.* Sensitivity analyses of the noise level $\xi$. BwOUCB does not use the executing probability $E$, its performance is independent of $\xi$. We can see that the performance of BwOUCB became close to or better than the other methods as $\xi$ increased.

For each $\boldsymbol{x}_n \in \tilde{\mathcal{X}}$, we set $P(Y = 1 \mid X = \boldsymbol{x}_n) = \max(\min(y_n + 0.1 \cdot \varepsilon, 1.0), 0.0)$, where $\varepsilon \sim \mathcal{N}(0, 1)$.

4. We obtained a sequence of instances $\boldsymbol{x}_1, \ldots, \boldsymbol{x}_T$ with $T = 1000$ by randomly sampling the instances $\boldsymbol{x}_n$ in the test set $S_{\text{te}}$ such that $h(\boldsymbol{x}_n) = 0$ and $y_n = 0$. Then, we started the procedure of Problem 3.1.

We repeated the above procedures 10 times. We used the $\ell_1$-norm $\|\boldsymbol{a}\|_1$ as the cost function $c$ and set $\nu = 1/D$ for computing the executing probability $E$. In each round $t \in [T]$, all the methods receive the instance $\boldsymbol{x}_t$ and its candidate actions $\mathcal{A}_t$ constructed by (3). In addition, the methods other than BwOUCB receive the execution probability $E(\boldsymbol{a} \mid \boldsymbol{x}_t)$ for each $\boldsymbol{a} \in \mathcal{A}_t$. As the delay distribution $\mathcal{D}$, we used the geometric distribution with the parameter $0.2$. We measured the average of (i) *improvement* (i.e., $h^*$-validity), (ii) *cost* $c(\boldsymbol{a}_t \mid \boldsymbol{x}_t)$, and (iii) *mean expected reward (MER)* $\mathcal{R}(t)$ in each round $t$. Due to the page limitation, we report the results on RF here, and those on NN in Appendix B.

**Baseline and Our Algorithm.** To the best of our knowledge, there is no existing AR method that ensures improvement from the long-term perspective. Thus, we compare our algorithms (*LinUCB* and *BwOUCB*) with two existing methods as baselines. One baseline is a method that max-

imizes $E$ over $\mathcal{A}_t$, which is equivalent to the standard AR method that solves the problem (1). By the definition of $\mathcal{A}_t$, it can be regarded as the existing AR method based on the class prototypes (*ProtoAR*) (Van Looveren & Klaise, 2021). Another baseline is the trustworthy actionable perturbation (*TAP*) (Friedbaum et al., 2024) that solves (1) under the constraint on trustworthiness, which is the true label probability of $\boldsymbol{x} + \boldsymbol{a}$. The trustworthiness is estimated using a verifier that is trained to predict whether a given instance is an adversarial example. For both LinUCB and BwOUCB, we set $m = 10$. We also set $\lambda = 20.0$ for LinUCB and $B = 50$ for BwOUCB, respectively. The sensitivity analyses of these parameters are shown in Appendix B.

### 6.2. Comparison under Noiseless Cost Evaluation

First, we evaluate the improvement and cost of actions obtained by each method. Figure 2(a) presents the results on the average improvement and cost. From Figure 2(a), we observe that (i) both LinUCB and BwOUCB stably outperformed the baselines in terms of the average improvement on all the datasets; (ii) LinUCB attained comparable average cost to the baselines. These results indicate that *our methods could suggest actions that stably achieve higher improvement than the baselines*, and *our LinUCB achieved higher improvement while maintaining comparable costs*.

Figure 2(b) shows the average mean expected reward of each round. From Figure 2(b), we see that the average mean expected rewards of LinUCB and BwOUCB increased as the number of rounds progressed, which are consistent with Proposition 4.2. Therefore, we confirmed that *our methods became to suggest better actions in the sense of their improvement and cost as the round progressed*.

### 6.3. Comparison under Noisy Cost Evaluation

Next, we examine each method under the noisy cost evaluation situation. To simulate the uncertainty in the cost evaluation, we added a noise $\xi \cdot \varepsilon$ to the execution probability $E(\boldsymbol{a} \mid \boldsymbol{x}_t)$ and scaled it in $[0, 1]$, where $\xi \geq 0$ is a noise level parameter and $\varepsilon \sim \mathcal{N}(0, 1)$. Such noisy information on $E$ was passed to methods other than BwOUCB. Figure 3 presents the results with the noise level $\xi = 0.25$. From Figure 3(a), we can see that both LinUCB and BwOUCB tended to achieve higher average improvement than the baselines, as with the results shown in Figure 2(a). On the other hand, the gap between the average cost of BwOUCB and the others decreased compared to the results under the noiseless situation. Furthermore, from Figure 3(b), we observe that BwOUCB achieved higher average mean expected rewards than LinUCB on all the datasets in the final round.

Finally, we analyze the sensitivity of the performance of each method to the noise level $\xi$. Figure 4 shows the results of the sensitivity analyses of the average improvement, cost,

and mean expected reward in the final round by varying $\xi$. Note that since BwOUCB does not use the executing probability $E$, its performance is independent of the noise level $\xi$. From Figures 4(a) and 4(b), we can see that the average improvement and cost of LinUCB were worse than or close to those of BwOUCB as $\xi$ increased. In addition, from Figure 4(c), we observed that the average mean expected reward of LinUCB became lower than BwOUCB as $\xi$ increased. In summary, we confirmed that *the performance of BwOUCB was close to or better than that of LinUCB in the situation where the cost evaluation includes uncertainty.*

# 7. Conclusion

This paper proposed algorithmic recourse for long-term improvement (ARLIM), a new framework for providing recourse actions that not only alter the undesired prediction results but also improve the real-world outcomes. We introduced a new problem setting in which instances arrive one by one, and the agent suggests an action for each instance and later observes delayed feedback on its outcome. By exploiting such feedback, we aimed to train an agent to suggest an action that improves the real-world outcome for each instance. We formulated our task as an online learning problem and proposed two practical algorithms based on contextual linear bandit and contextual Bayesian optimization. Experimental results demonstrated the efficacy of our methods in comparison to the existing baselines.

**Limitations and Future Work**

Our framework has some limitations that should be addressed in future work. One major limitation is the inherent challenges posed by the long-term nature of recourse implementation. There exist real-world applications where it may take several years until we observe the feedback on the outcomes of suggested recourse actions. Such situations correspond to the case where the delay variable $D_t \sim \mathcal{D}$ of each round $t$ is large. Because such a long-delayed feedback may lead to a lack of sufficient data for training the agent, it can be problematic for our framework. Note that we are the first to demonstrate the feasibility of suggesting improvement-oriented actions by exploiting feedback, even when only delayed feedback is available. Even if it takes a long time to observe the first feedback, our framework is expected to work well once it starts to observe feedback. However, to ensure improvement-oriented actions for instances in the early rounds, we need to address long delays (e.g., by exploiting intermediate observations (Vernade et al., 2020b; Esposito et al., 2023)), but such an extension of our framework is non-trivial and remains a challenge.

In addition, there are several interesting directions to make our framework more practical. First, while we assume that a classifier $h$ is fixed over all the rounds, it is often updated over time (Upadhyay et al., 2021). While we empirically confirmed that our methods work better than the baselines even in the scenario where the classifier is frequently updated in Appendix B.5, extending our framework so as to adaptively handle such a non-stationary setting is interesting. Second, our framework implicitly assumes that users execute their suggested actions completely if they accept to execute the actions. In practice, users may execute the actions partially or incorrectly (Pawelczyk et al., 2023), which can lead to a gap between the observed feedback and the true outcomes. Finally, deriving better theoretical guarantees for our algorithms is important for practical applications. While we show a theoretical bound of Algorithm 1 in Proposition 4.2, it depends on the total number of arms $K$, which is weak in the literature of CLB (Vernade et al., 2020a). It is also important to derive a theoretical guarantee for Algorithm 2 by extending the existing regret bound of CBO (Verma et al., 2022). Furthermore, since our model and algorithm are direct applications of the previous studies on CLB and CBO, it is an interesting direction for future research to design more suited models and algorithms for Problem 3.1 (Bouneffouf et al., 2020; Wang et al., 2023).

# Acknowledgments

We wish to thank Yuichi Ike and Shion Takeno for making a number of valuable suggestions. We also thank the anonymous reviewers for their insightful comments. This work was supported in part by JST ACT-X JPMJAX23C6, JSPS KAKENHI Grant-in-Aid for Early-Career Scientists 24K17465, Grant-in-Aid for Scientific Research(B) 23K28146, and IMI Joint Usage/Research Center in Kyushu University (FY2024 Short-term Joint Research 2024a032).

# Impact Statement

Our proposed method, algorithmic recourse for long-term improvement (ARLIM), is a new framework for providing recourse actions that not only alter the undesired predictions but also improve real-world outcomes. As mentioned in the paper, this framework can lead to more reliable and beneficial decision-making processes by suggesting recourse actions that ensure improvement for as many users as possible. Our focus on long-term improvement can lead to more sustainable and positive outcomes for users, which is beneficial for both users and decision-makers. On the other hand, our framework also has potential societal impacts that need careful consideration. For example, our methods rely on the availability of feedback that tracks users over time, which requires us to carefully consider its privacy risk. Overall, the proposed method has the potential to significantly improve the effectiveness and reliability of the decision-making process, but we need careful consideration of its risks before incorporating it into the actual decision-making process.

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

## A. Omitted Proofs

### A.1. Proof of Proposition 4.1

We show that Problem 3.1 is an instance of the contextual linear bandit problem under stochastic delayed feedback defined as follows (Vernade et al., 2020a).

**Problem A.1** (Contextual Linear Bandit Problem under Stochastic Delayed Feedback (Vernade et al., 2020a)). We assume a number of rounds $T \in \mathbb{N}$ and unknown parameter $\boldsymbol{\theta} \in \mathbb{R}^P$ with $\|\theta^*\|_2 \leq 1$. For each round $t \in [T]$, the following procedure is repeated between an agent and environment:

1. The agent receives a set of $K_t$ candidate arms $\mathcal{V}_t \subset \mathbb{R}^P$ with $\boldsymbol{v}^\top \boldsymbol{\theta} \in [0, 1]$ and $\|\boldsymbol{v}\|_2 \leq 1$ for any $\boldsymbol{v} \in \mathcal{V}_t$.

2. The agent selects an arm $V_t \in \mathcal{V}_t$ based on past observations and sends it to the environment.

3. The environment samples a reward $R_t \in \{0, 1\}$ and delay $D_t \in \mathbb{N}$ from $\mathcal{B}(V_t^\top \boldsymbol{\theta})$ and $\mathcal{D}$, respectively.

4. The agent observes a set of the past rewards $\{R_s \mid s + D_s = t\}_{s=1}^{t-1}$ that becomes observable at round $t$.

The goal of the agent is to minimize the cumulative regret $\text{regret}(T) = \sum_{t=1}^{T} (\boldsymbol{\theta}^\top V_t^* - \boldsymbol{\theta}^\top V_t)$, where $V_t^* = \arg\max_{\boldsymbol{v} \in \mathcal{V}_t} \boldsymbol{\theta}^\top \boldsymbol{v}$.

*Proof of Proposition 4.1.* For Problem 3.1, we denote $\mathcal{A}_t = \{\boldsymbol{a}_1, \ldots, \boldsymbol{a}_L\}$. Recall that there exists a mapping $\pi_t \colon [L] \to [K]$ such that $\boldsymbol{x}_t + \boldsymbol{a}_l = \tilde{\boldsymbol{x}}_{\pi_t(l)}$ for any $l \in [L]$ by definition. Let $\phi_t(\boldsymbol{a}_l) = E(\boldsymbol{a}_l \mid \boldsymbol{x}_t) \cdot \boldsymbol{e}_{\pi_t(l)}$, where $\boldsymbol{e}_k \in \{0, 1\}^K$ is the binary vector where its $k$-th element is 1 and the others are 0. By definition, we have $I(\boldsymbol{a}_l \mid \boldsymbol{x}_t) = P(Y = 1 \mid X = \tilde{\boldsymbol{x}}_{\pi_t(l)})$. We denote $\boldsymbol{\theta} = (P(Y = 1 \mid X = \tilde{\boldsymbol{x}}_1), \ldots, P(Y = 1 \mid X = \tilde{\boldsymbol{x}}_K))$ and $A_t = \phi_t(\boldsymbol{a}_t)$ for $\boldsymbol{a}_t$ in Problem 3.1. Then, we have $A_t^\top \boldsymbol{\theta} = E(\boldsymbol{a}_t \mid \boldsymbol{x}_t) \cdot I(\boldsymbol{a}_t \mid \boldsymbol{x}_t)$. Therefore, our reward $R_t$ of the round $t$ can be expressed as $R_t \sim \mathcal{B}(A_t^\top \boldsymbol{\theta})$. Furthermore, $\phi_t(\boldsymbol{a})^\top \boldsymbol{\theta} \in [0, 1]$ and $\|\phi_t(\boldsymbol{a})\|_2 \leq 1$ hold for any $\boldsymbol{a} \in \mathcal{A}_t$. Finally, minimizing our mean expected reward $\mathcal{R}(T)$ is equivalent to maximizing $\sum_{t=1}^{T} (\boldsymbol{\theta}^\top A_t^* - \boldsymbol{\theta}^\top A_t)$, where $A_t^* = \arg\max_{\boldsymbol{a}_l \in \mathcal{A}_t} \boldsymbol{\theta}^\top \phi_t(\boldsymbol{a}_l)$. In summary, we can see that Problem 3.1 is reduced to Problem A.1 by replacing (i) $\mathcal{V}_t$ with $\{\phi_t(\boldsymbol{a}_1), \ldots, \phi_t(\boldsymbol{a}_L)\}$; (ii) $V_t$ with $\phi_t(\boldsymbol{a}_t)$; (iii) $\boldsymbol{\theta}$ with $(P(Y = 1 \mid X = \tilde{\boldsymbol{x}}_1), \ldots, P(Y = 1 \mid X = \tilde{\boldsymbol{x}}_K))$, respectively. $\square$

### A.2. Proof of Proposition 4.2

*Proof.* By definition, $\mathbb{E}[R_t] = E(\boldsymbol{a}_t \mid \boldsymbol{x}_t) \cdot I(\boldsymbol{a}_t \mid \boldsymbol{x}_t)$ holds. Let $R_t^* = \max_{\boldsymbol{a} \in \mathcal{A}_t} E(\boldsymbol{a} \mid \boldsymbol{x}_t) \cdot I(\boldsymbol{a} \mid \boldsymbol{x}_t)$ be the maximum expected reward in a round $t$. Then, by applying Theorem 2 of (Vernade et al., 2020a), we have

$$\sum_{t=1}^{T} (R_t^* - \mathbb{E}[R_t]) \leq \frac{4}{\tau_m} \cdot \left( \alpha_T \sqrt{2KT \log\left(\frac{K\lambda + T}{K\lambda}\right)} + mK \log\left(\frac{K\lambda + T}{K\lambda}\right) \right)$$

$$\iff \frac{1}{T} \sum_{t=1}^{T} R_t^* - \frac{1}{T} \sum_{t=1}^{T} \mathbb{E}[R_t] \leq \frac{4}{\tau_m} \cdot \frac{1}{T} \cdot \left( \alpha_T \sqrt{2K \log\left(\frac{K\lambda + T}{K\lambda}\right)} + mK \log\left(\frac{K\lambda + T}{K\lambda}\right) \right)$$

$$\iff \mathcal{R}_T^* - \mathcal{R}(T) \leq \frac{4}{\tau_m} \cdot \Gamma_T$$

$$\iff \mathcal{R}(T) \geq \mathcal{R}_T^* - \frac{4}{\tau_m} \cdot \Gamma_T,$$

which concludes the proof. $\square$

## B. Additional Experimental Results

### B.1. Details of Datasets and Preprocessing

Table 1 presents the details on the value type, minimum value, mean value, maximum value, and immutability of each feature of the datasets that we used in our experiments. For numerical immutable features such as age in each dataset, we transformed them into binary features by comparing them with their medians. All the categorical features were transformed into binary features by one-hot encoding. All the numerical features were normalized to $[0, 1]$.

### B.2. Complete Experimental Results of Section 6

Figures 5 to 8 present the complete experimental results of the baseline comparison for random forests (RF) and two-layer neural networks (NN) under the noiseless and noisy cost evaluation situations, respectively. Figures 9 and 10 show the sensitivity analyses of the noise level $\xi$ for RF and NN, respectively.

### B.3. Problem Settings beyond Assumptions

Here, we examine our methods under the scenarios beyond the assumptions of Problem 3.1. First, we consider the situation where the cost function $c$ that each method knows is different from that is used to compute the reward. In this case, each method receives the executing probability $E$ computed by one cost function $c$ and the reward $R_t$ computed by another cost function $c'$. We used the $\ell_1$-norm as $c$ and the max percentile shift (Ustun et al., 2019) as $c'$. Figures 11 to 14 present the results of the cost-mixture scenario. We can see that the performance of BwOUCB was close to or better than that of LinUCB, even in the noiseless cost evaluation situation.

Second, we consider the situation where the delay $D_t$ depends on the instance $\boldsymbol{x}_t$. In this case, we sample $D_t$ from the geometric distribution with the parameter $p = 0.2 \cdot x_{t,d} + 0.05 \cdot (1 - x_{t,d})$, where $x_{d,i} \in \{0, 1\}$ is the feature value of $\boldsymbol{x}_t$ that indicates whether its age is younger than the median of the dataset or not. That is, the delay variable $D_t$ is sampled from different distributions depending on the age of the instance $\boldsymbol{x}_t$. Figures 15 and 16 present the results of the adaptive delay scenario. We observed that LinUCB and BwOUCB performed similarly to the results under the delay distribution that is independent of the instance shown in Figures 5 and 7.

### B.4. Sensitivity Analyses of Parameters

Figures 17 and 18 show the sensitivity analyses of the total number of recourse instances $K$. Figures 19 and 20 present the sensitivity analyses of the window parameter $m$. Figures 21 to 24 show the sensitivity analyses of the regularization parameter $\lambda$ of LinUCB. Figures 25 and 26 present the sensitivity analyses of the number of trees $B$ of BwOUCB.

### B.5. Comparison under Non-stationary Setting

We examine the performance of each method under a non-stationary setting where the classifier $h$ is updated over time. To simulate this situation, we updated the classifier $h$ using the set of the past instance-reward pairs $(\boldsymbol{x}_s + \boldsymbol{a}_s, R_s)$ that become observable until a round $t > s$. By varying the frequency of the update, we examined the performance of each method. Figures 27 and 28 present the results of the model update scenario where the update frequency is once every 25 rounds. Figures 29 and 30 show the sensitivity analyses of the update frequency. We can see that the performance of each method was not significantly affected by the update frequency and that these results were not so different from those of the stationary setting.

### B.6. Comparison to Algorithm Based on Gaussian Process

We compare our methods with the existing algorithm based on the Gaussian process (*CGPUCB*) (Verma et al., 2022) under the same setting with Figure 2. For CGPUCB, we used the radial basis function (RBF) kernel and optimized the kernel parameters once every 10 rounds. Figures 31 and 32 present the results of the comparison. We observed that the performance of CGPUCB was close to or worse than that of BwOUCB. Table 2 shows the average running time of each method in each round. We can see that the running time of CGPUCB was significantly longer than that of BwOUCB. In summary, we confirmed that *the performance of BwOUCB was close to or better than that of CGPUCB* and *BwOUCB was significantly faster than CGPUCB*.

## C. Additional Comments on Existing Assets

All the code used in our experiments was implemented in Python 3.10 with scikit-learn 1.5.2. Scikit-learn 1.5.2 is publicly available under the BSD-3-Clause license. All the scripts and datasets are available in our GitHub repository at https://github.com/kelicht/arlim. All the datasets used in our experiments are publicly available and do not contain any identifiable information or offensive content. As they are accompanied by appropriate citations in the main body, see the corresponding references for more details. All the experiments were conducted on macOS Sequoia with Apple M2 Ultra CPU and 128 GB memory.

*Table 1.* Details of the datasets used in the experiments.

(a) Credit (Yeh & hui Lien, 2009)

| Feature | Type | Min | Mean | Max | Immutable |
|---|---|---|---|---|---|
| Married | Binary | 0 | 0.455300 | 1 | Yes |
| Single | Binary | 0 | 0.532133 | 1 | Yes |
| Age_lt_40 | Binary | 0 | 0.724200 | 1 | Yes |
| EducationLevel | Numerical | 0 | 2.157733 | 3 | No |
| MaxBillAmountOverLast6Months | Numerical | 0 | 1849.565000 | 50810 | No |
| MaxPaymentAmountOverLast6Months | Numerical | 0 | 483.785333 | 51430 | No |
| MonthsWithZeroBalanceOverLast6Months | Numerical | 0 | 0.788833 | 6 | No |
| MonthsWithLowSpendingOverLast6Months | Numerical | 0 | 2.833133 | 6 | No |
| MonthsWithHighSpendingOverLast6Months | Numerical | 0 | 1.208333 | 6 | No |
| MostRecentBillAmount | Numerical | 0 | 1564.743000 | 29450 | No |
| MostRecentPaymentAmount | Numerical | 0 | 172.783000 | 26670 | No |
| TotalOverdueCounts | Numerical | 0 | 0.371600 | 3 | No |
| TotalMonthsOverdue | Numerical | 0 | 1.687700 | 36 | No |

(b) Diabetes (Dua & Graff, 2017)

| Feature | Type | Min | Mean | Max | Immutable |
|---|---|---|---|---|---|
| Pregnancies_lt_3 | Binary | 0.000 | 0.454427 | 1.00 | Yes |
| Glucose | Numerical | 44.000 | 121.681605 | 199.00 | No |
| BloodPressure | Numerical | 24.000 | 72.254807 | 122.00 | No |
| SkinThickness | Numerical | 0.000 | 20.536458 | 99.00 | No |
| Insulin | Numerical | 0.000 | 79.799479 | 846.00 | No |
| BMI | Numerical | 18.200 | 32.450805 | 67.10 | No |
| DiabetesPedigreeFunction | Numerical | 0.078 | 0.471876 | 2.42 | No |
| Age_gt_29 | Binary | 0.000 | 0.484375 | 1.00 | Yes |

(c) COMPAS (Angwin et al., 2016)

| Feature | Type | Min | Mean | Max | Immutable |
|---|---|---|---|---|---|
| juv_fel_count | Numerical | 0 | 0.059186 | 20 | No |
| juv_misd_count | Numerical | 0 | 0.091292 | 13 | No |
| juv_other_count | Numerical | 0 | 0.110751 | 9 | No |
| priors_count | Numerical | 0 | 3.247446 | 38 | No |
| c_charge_degree:F | Binary | 0 | 0.643100 | 1 | No |
| c_charge_degree:M | Binary | 0 | 0.356900 | 1 | No |
| age_lt_31 | Binary | 0 | 0.512567 | 1 | Yes |
| gender | Binary | 0 | 0.809794 | 1 | Yes |
| race | Binary | 0 | 0.514513 | 1 | Yes |

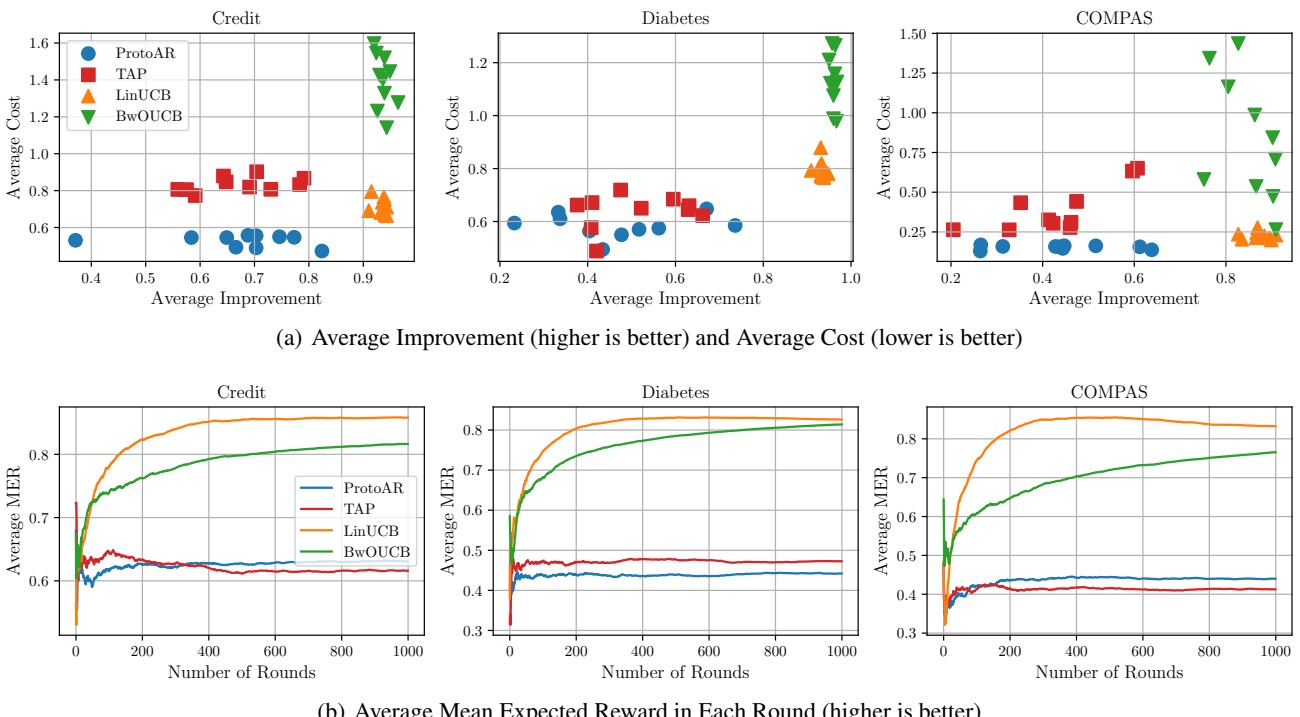

(a) Average Improvement (higher is better) and Average Cost (lower is better)

(b) Average Mean Expected Reward in Each Round (higher is better)

*Figure 5.* Experimental results of baseline comparison for RF under the noiseless cost evaluation situation.

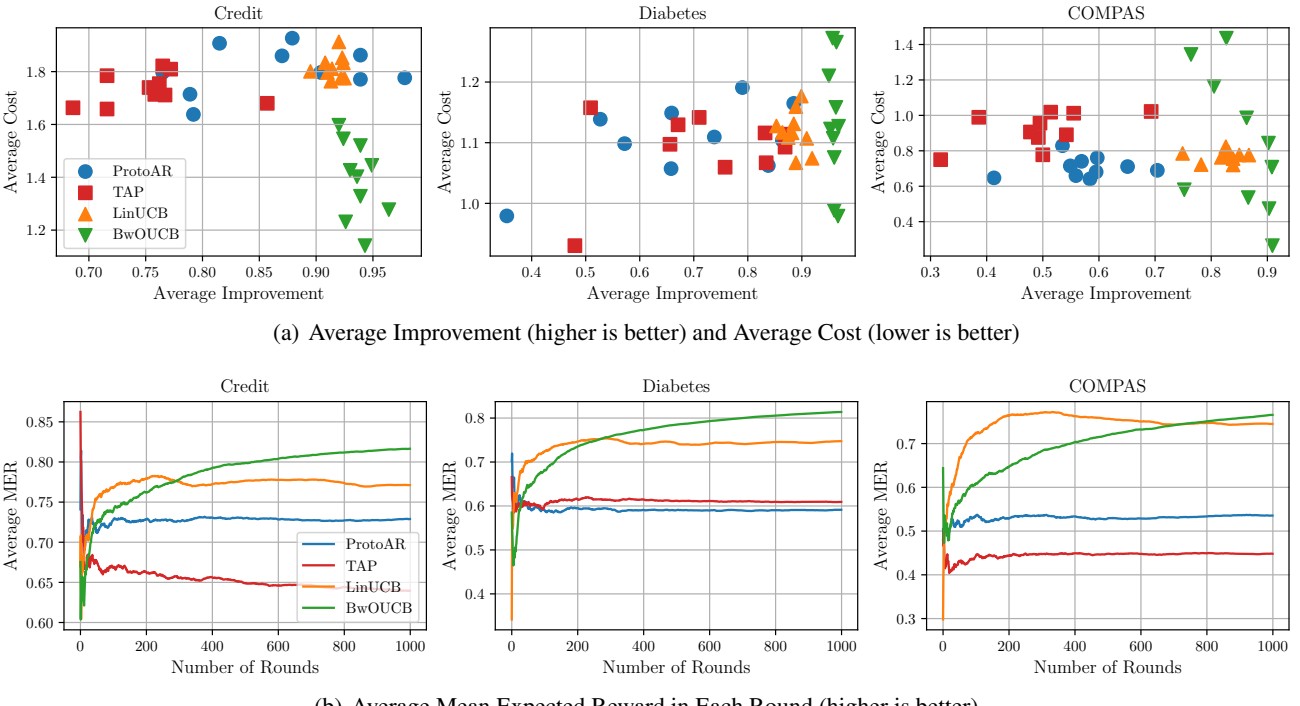

(a) Average Improvement (higher is better) and Average Cost (lower is better)

(b) Average Mean Expected Reward in Each Round (higher is better)

*Figure 6.* Experimental results of baseline comparison for RF under the noisy cost evaluation situation with the noise level $\xi = 0.25$.

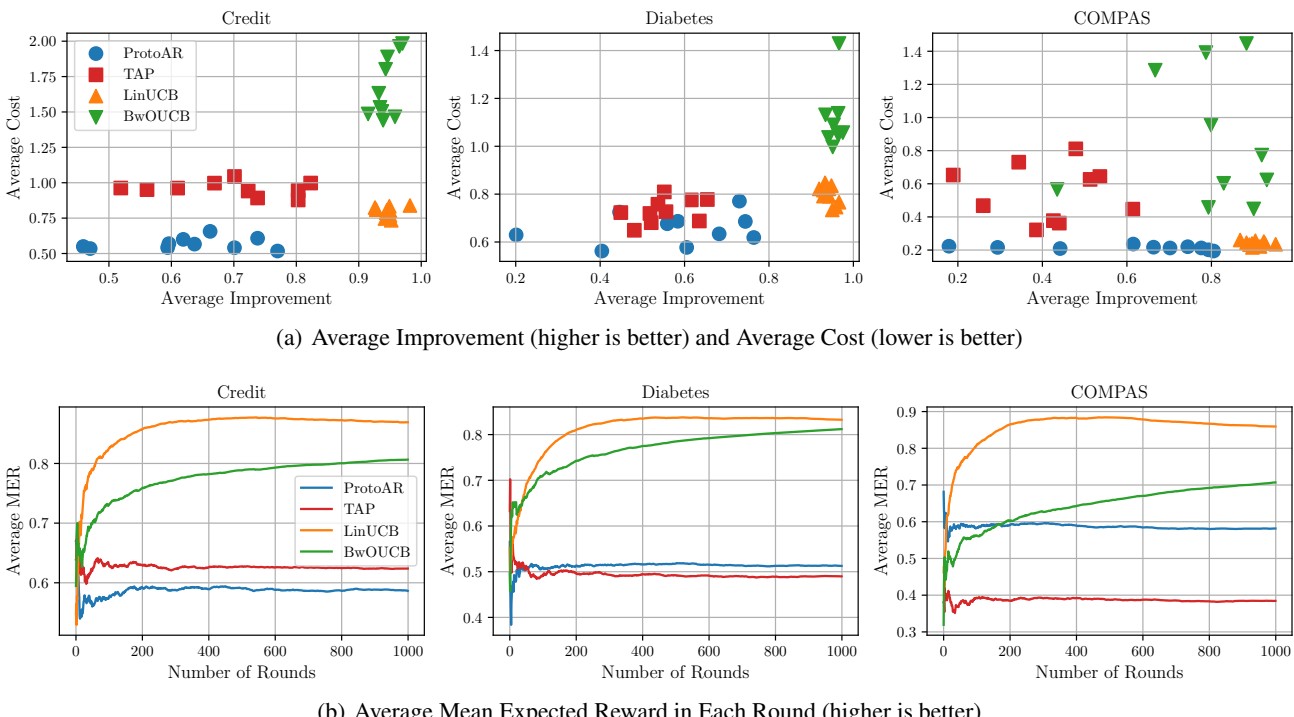

(a) Average Improvement (higher is better) and Average Cost (lower is better)

(b) Average Mean Expected Reward in Each Round (higher is better)

*Figure 7.* Experimental results of baseline comparison for NN under the noiseless cost evaluation situation.

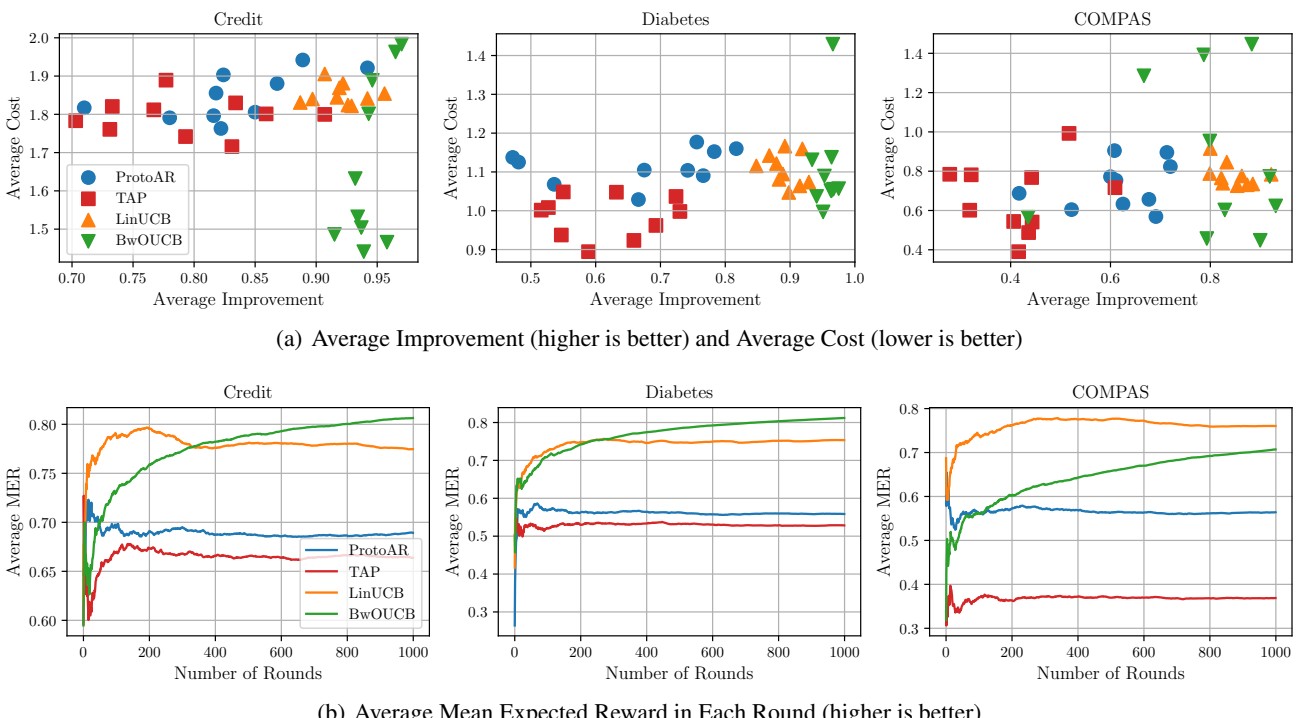

(a) Average Improvement (higher is better) and Average Cost (lower is better)

(b) Average Mean Expected Reward in Each Round (higher is better)

*Figure 8.* Experimental results of baseline comparison for NN under the noisy cost evaluation situation with the noise level $\xi = 0.25$.

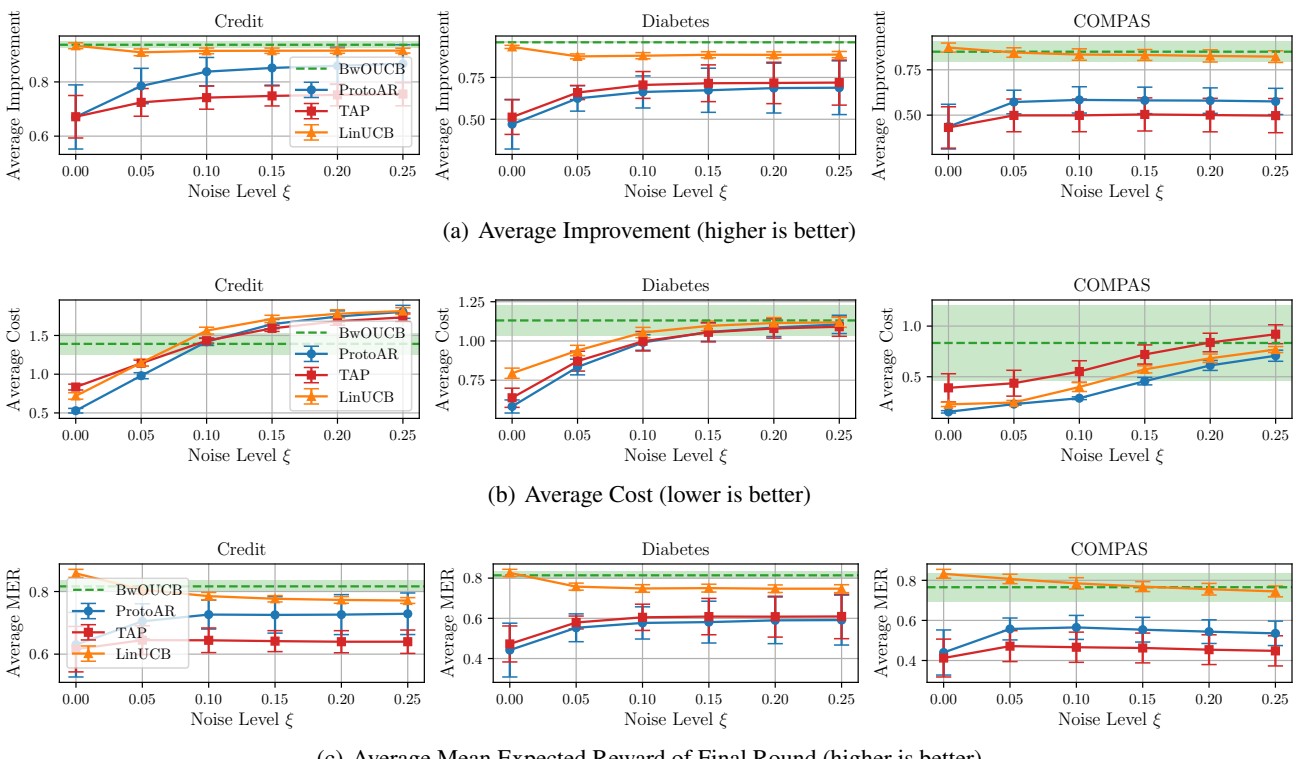

(a) Average Improvement (higher is better)

(b) Average Cost (lower is better)

(c) Average Mean Expected Reward of Final Round (higher is better)

*Figure 9.* Sensitivity analyses of the noise level $\xi$ for RF.

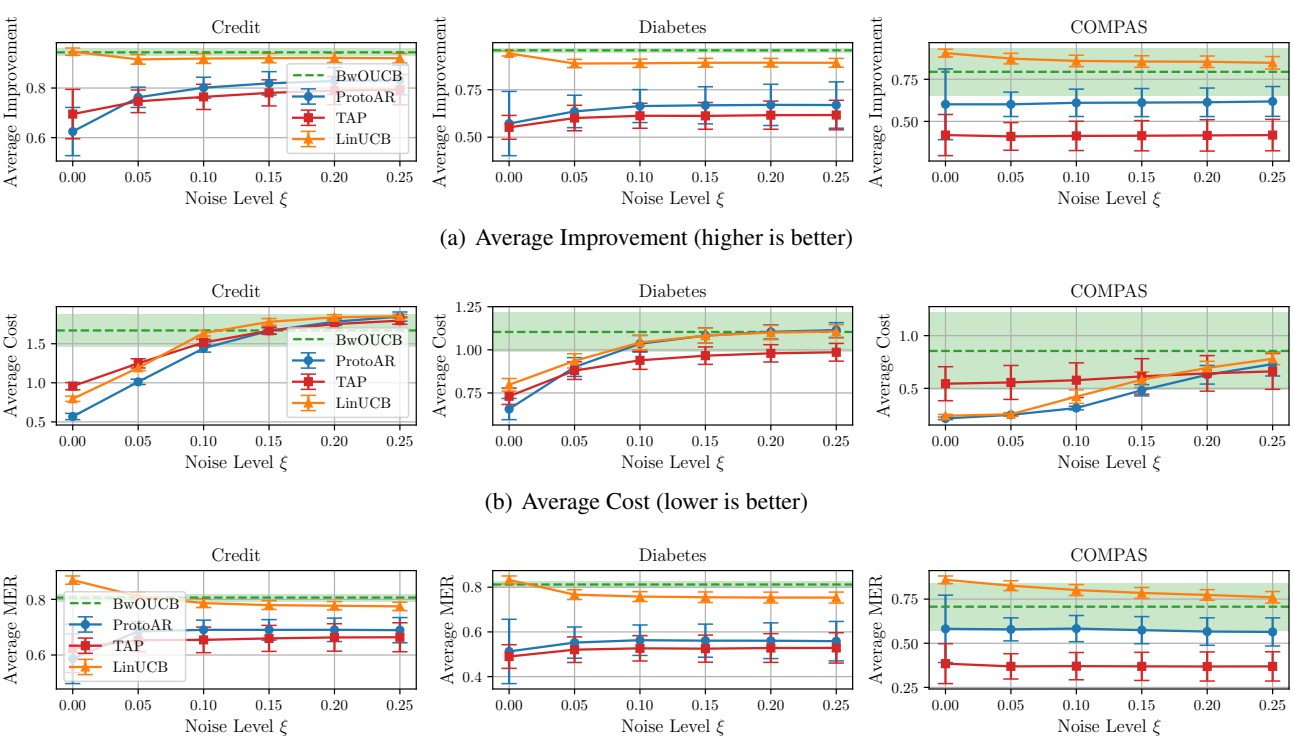

(a) Average Improvement (higher is better)

(b) Average Cost (lower is better)

(c) Average Mean Expected Reward of Final Round (higher is better)

*Figure 10.* Sensitivity analyses of the noise level $\xi$ for NN.

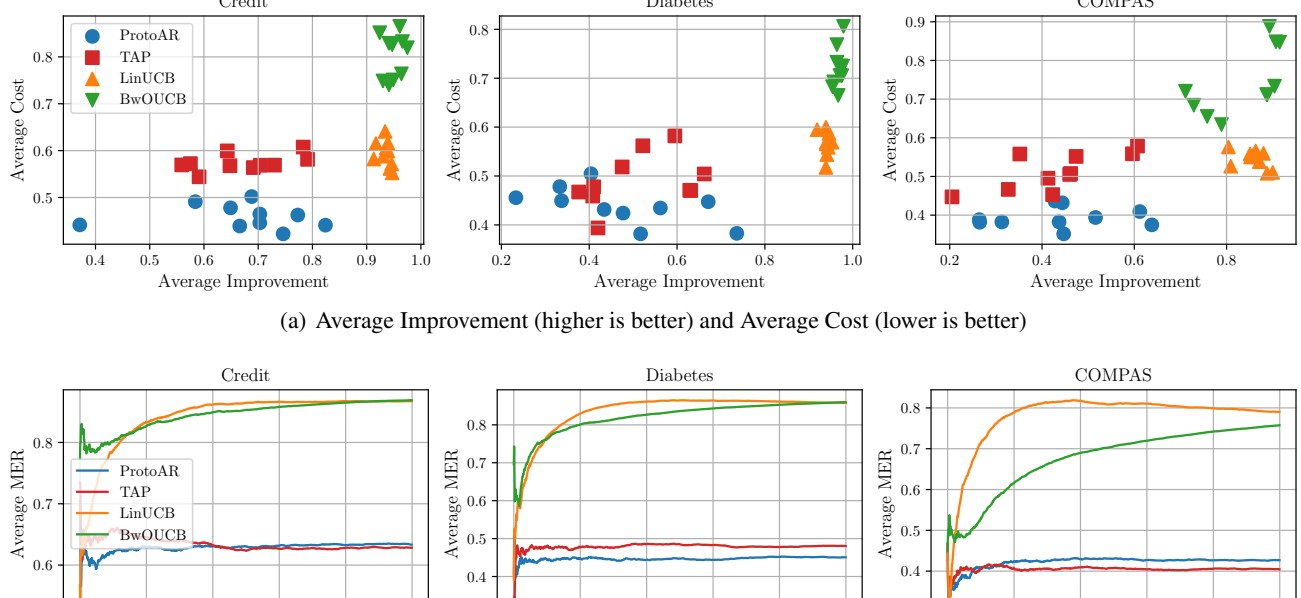

(a) Average Improvement (higher is better) and Average Cost (lower is better)

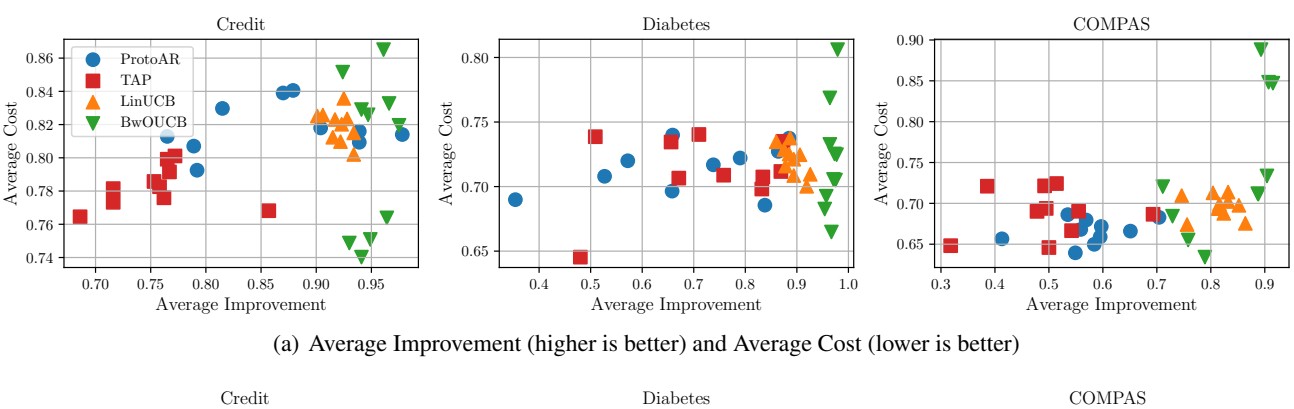

(b) Average Mean Expected Reward in Each Round (higher is better)

*Figure 11.* Experimental results of the cost-mixture scenario for RF under the noiseless cost evaluation situation.

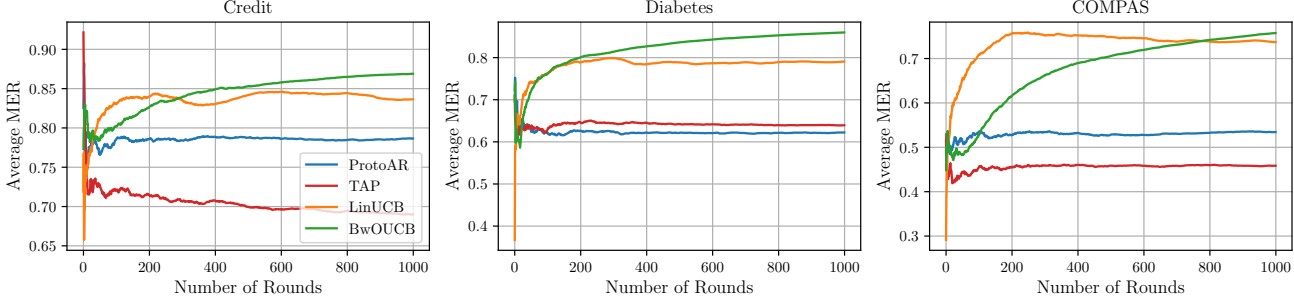

(a) Average Improvement (higher is better) and Average Cost (lower is better)

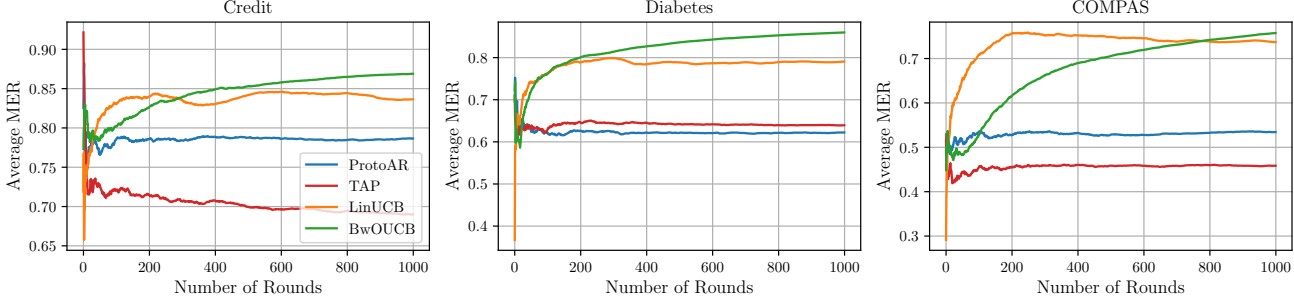

(b) Average Mean Expected Reward in Each Round (higher is better)

*Figure 12.* Experimental results of the cost-mixture scenario for RF under the noisy cost evaluation situation with the noise level $\xi = 0.25$.

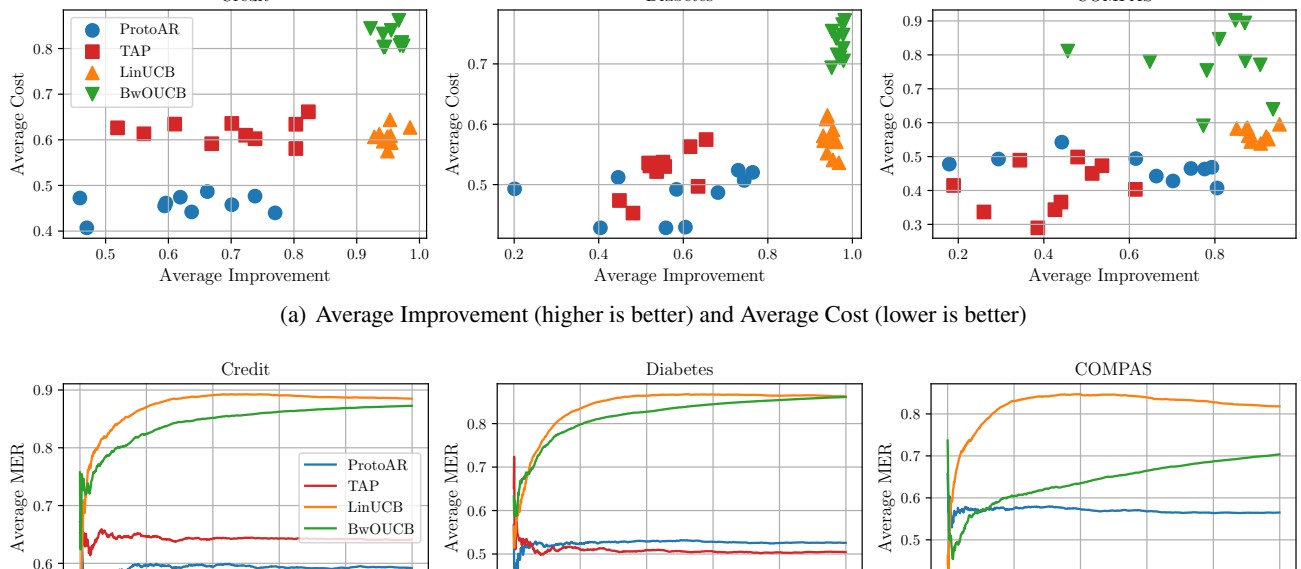

(a) Average Improvement (higher is better) and Average Cost (lower is better)

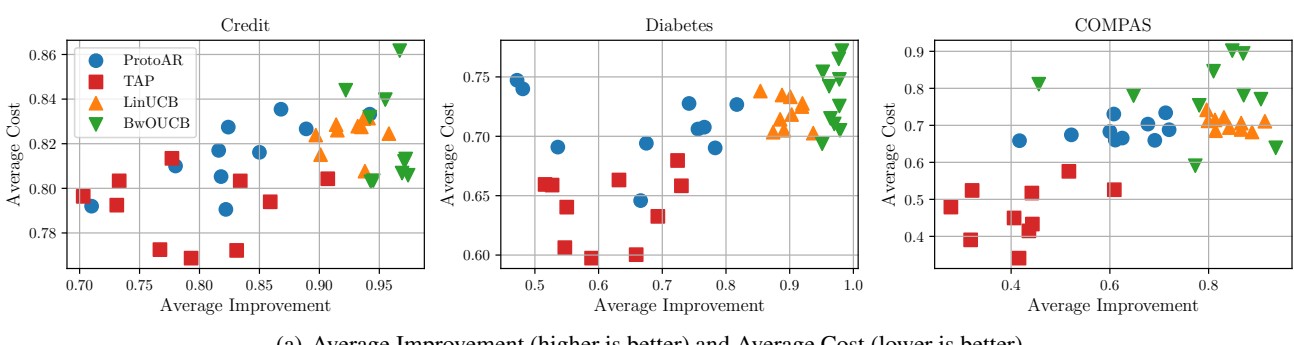

(b) Average Mean Expected Reward in Each Round (higher is better)

*Figure 13.* Experimental results of the cost-mixture scenario for NN under the noiseless cost evaluation situation.

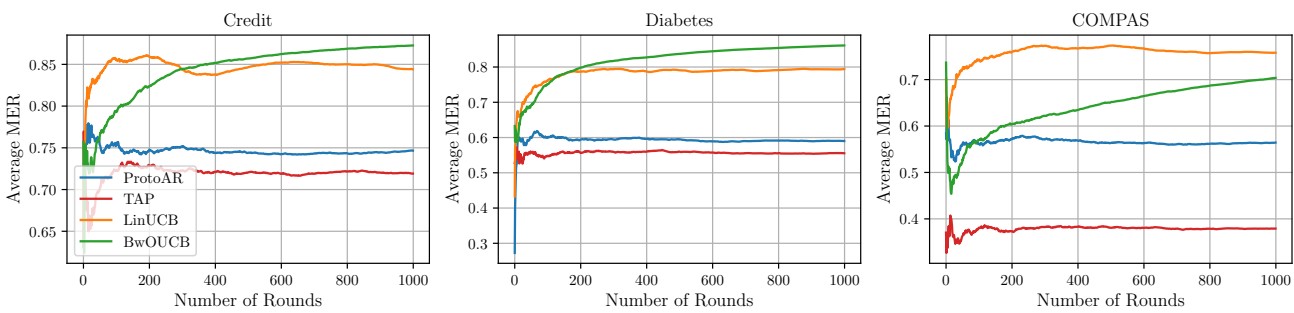

(a) Average Improvement (higher is better) and Average Cost (lower is better)

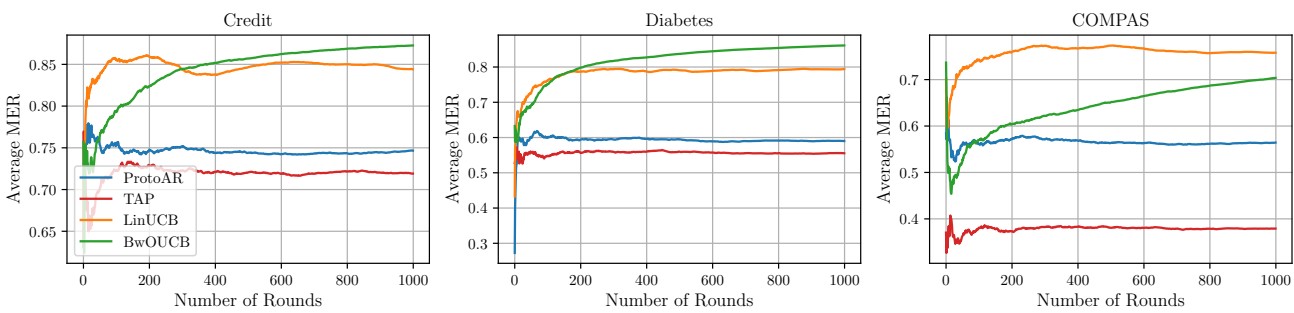

(b) Average Mean Expected Reward in Each Round (higher is better)

*Figure 14.* Experimental results of the cost-mixture scenario for NN under the noisy cost evaluation situation with the noise level $\xi = 0.25$.

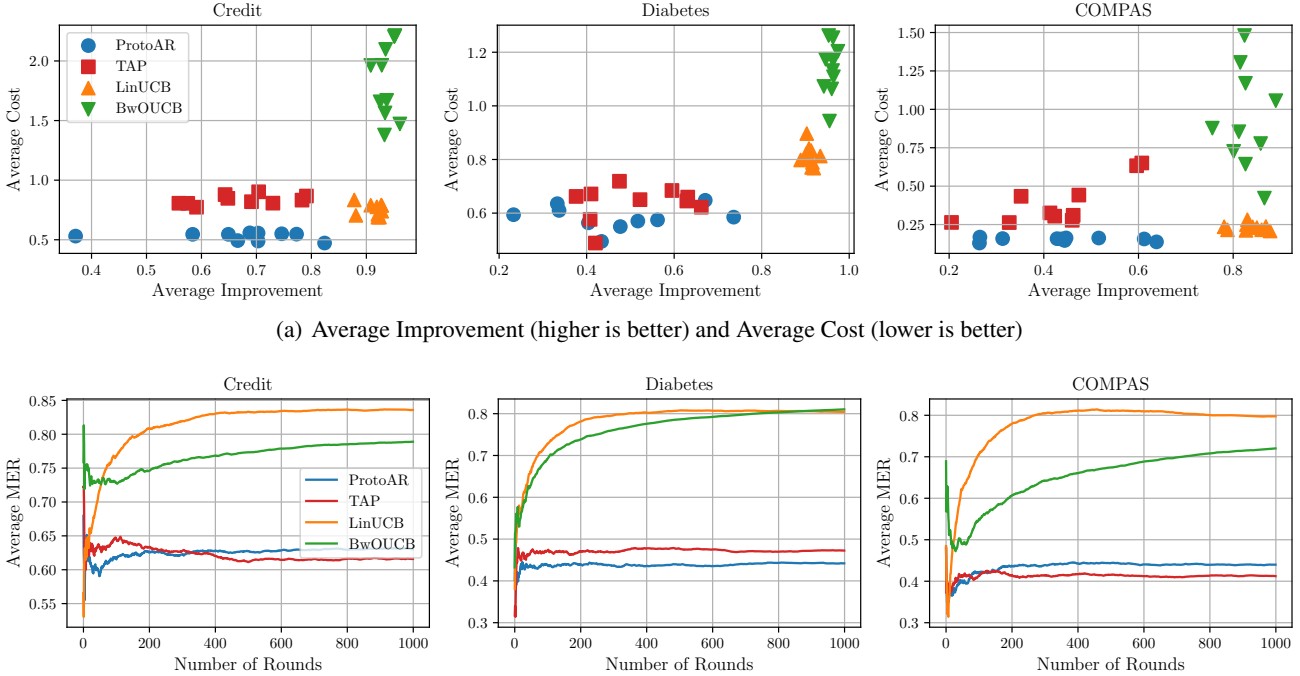

(a) Average Improvement (higher is better) and Average Cost (lower is better)

(b) Average Mean Expected Reward in Each Round (higher is better)

*Figure 15.* Experimental results of the adaptive delay scenario for RF.

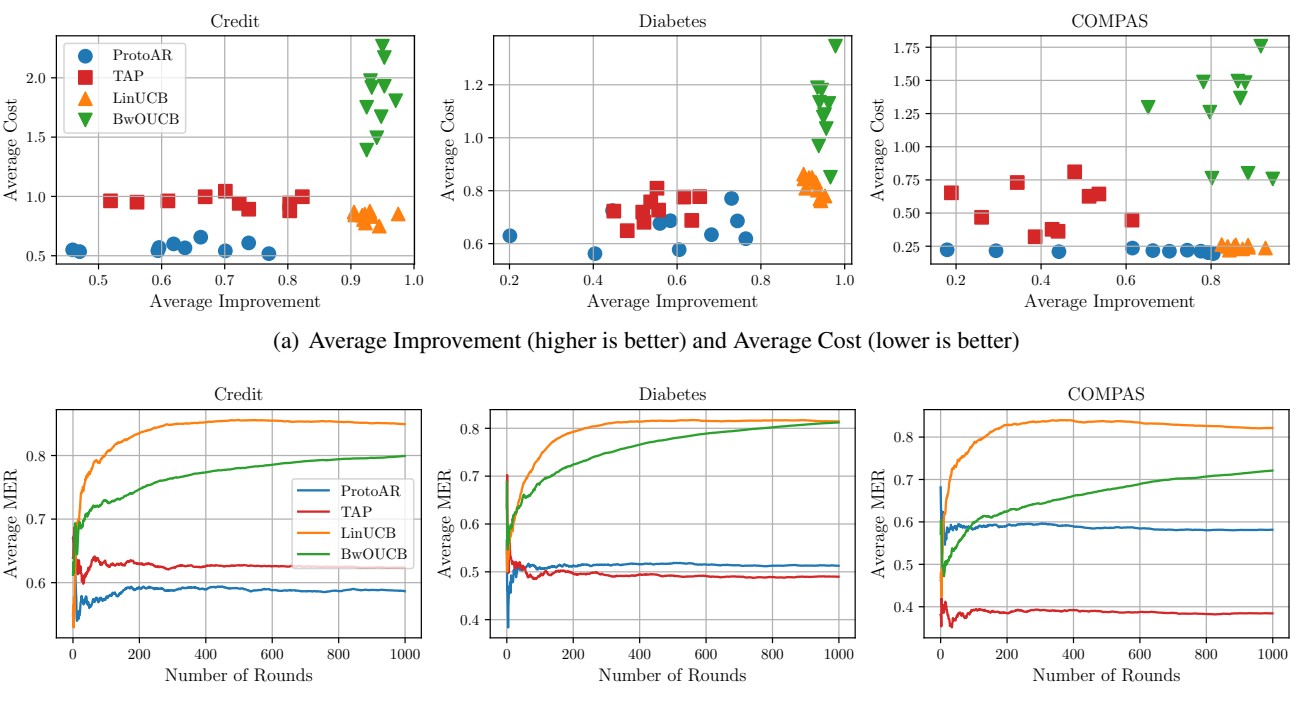

(a) Average Improvement (higher is better) and Average Cost (lower is better)

(b) Average Mean Expected Reward in Each Round (higher is better)

*Figure 16.* Experimental results of the adaptive delay scenario for NN.

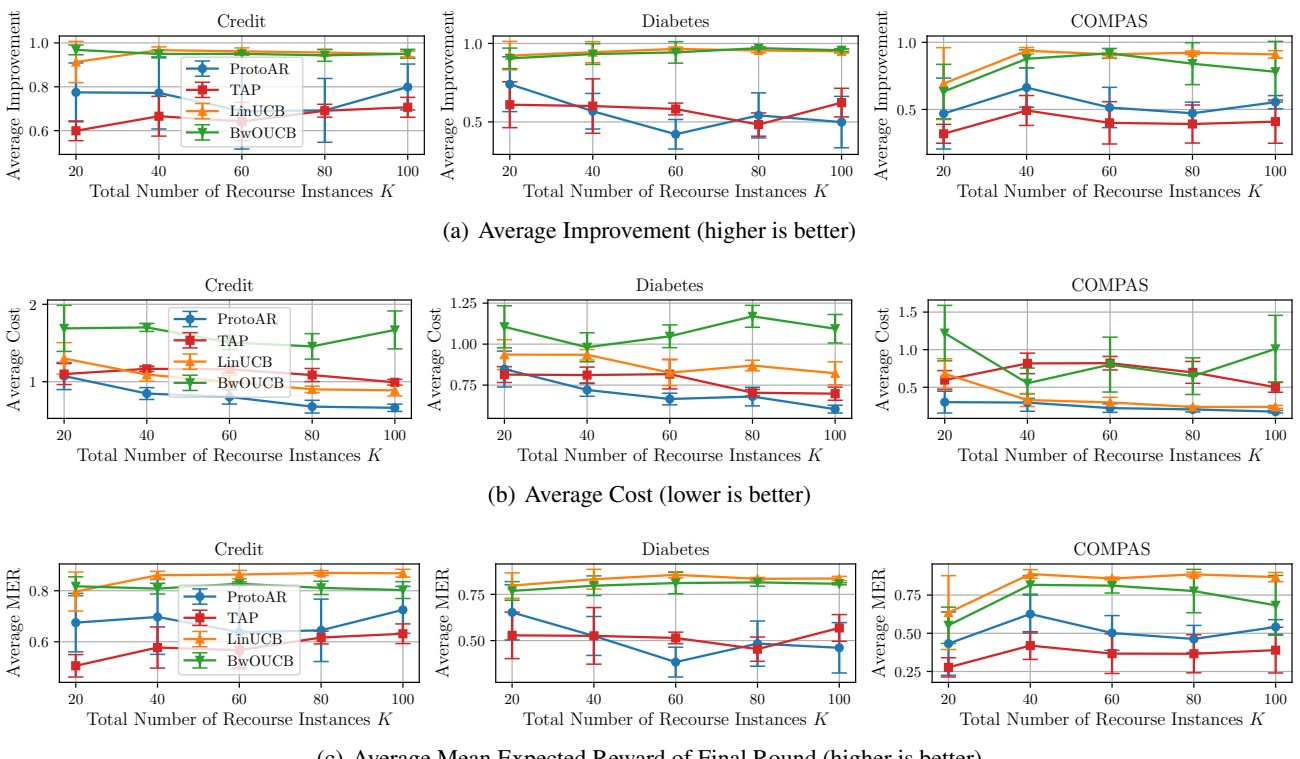

(a) Average Improvement (higher is better)

(b) Average Cost (lower is better)

(c) Average Mean Expected Reward of Final Round (higher is better)

*Figure 17.* Sensitivity analyses of the total number of recourse instances $K$ for RF.

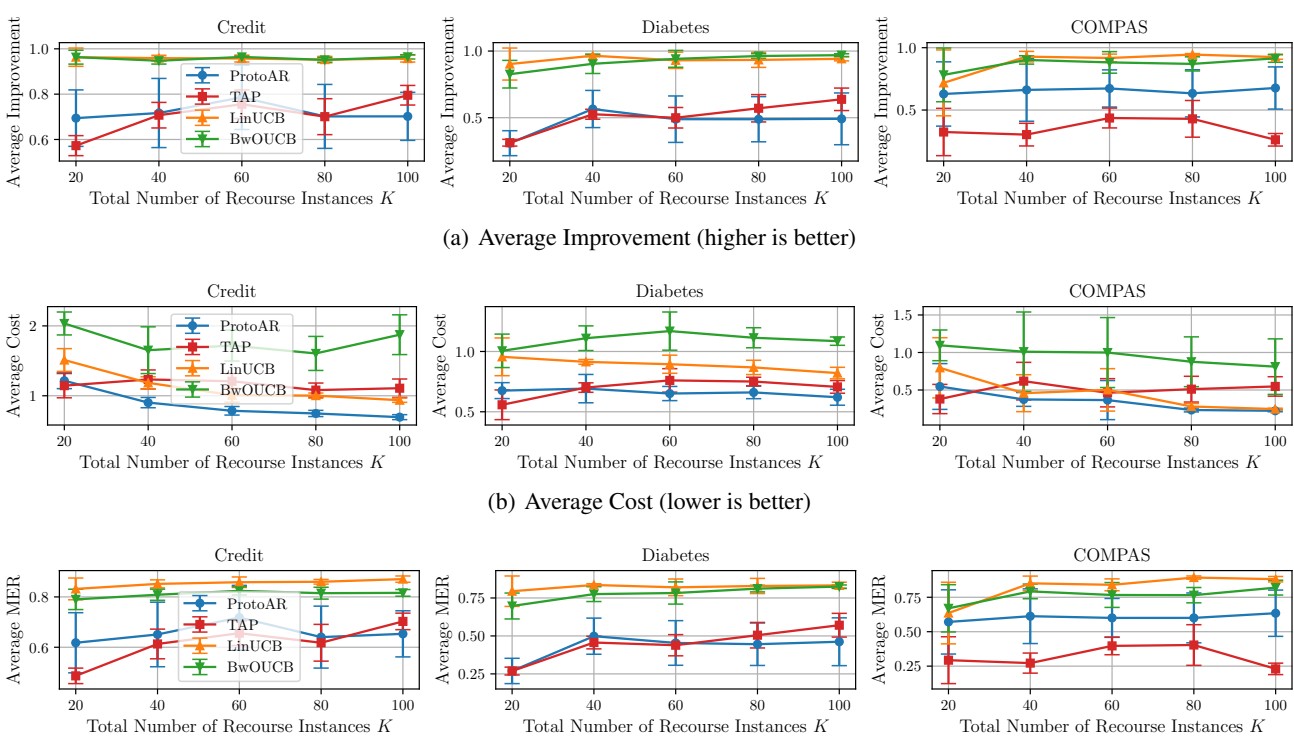

(a) Average Improvement (higher is better)

(b) Average Cost (lower is better)

(c) Average Mean Expected Reward of Final Round (higher is better)

*Figure 18.* Sensitivity analyses of the total number of recourse instances $K$ for NN.

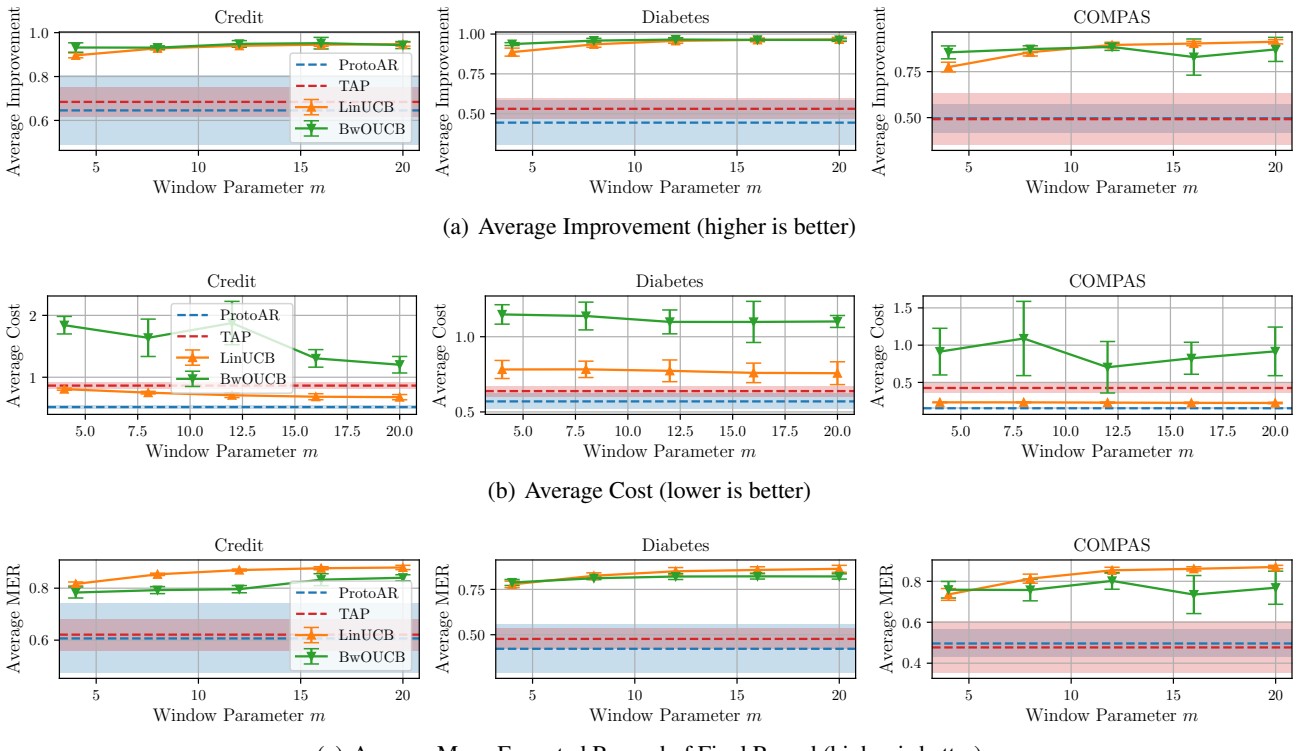

(a) Average Improvement (higher is better)

(b) Average Cost (lower is better)

(c) Average Mean Expected Reward of Final Round (higher is better)

*Figure 19.* Sensitivity analyses of the window parameter $m$ for RF.

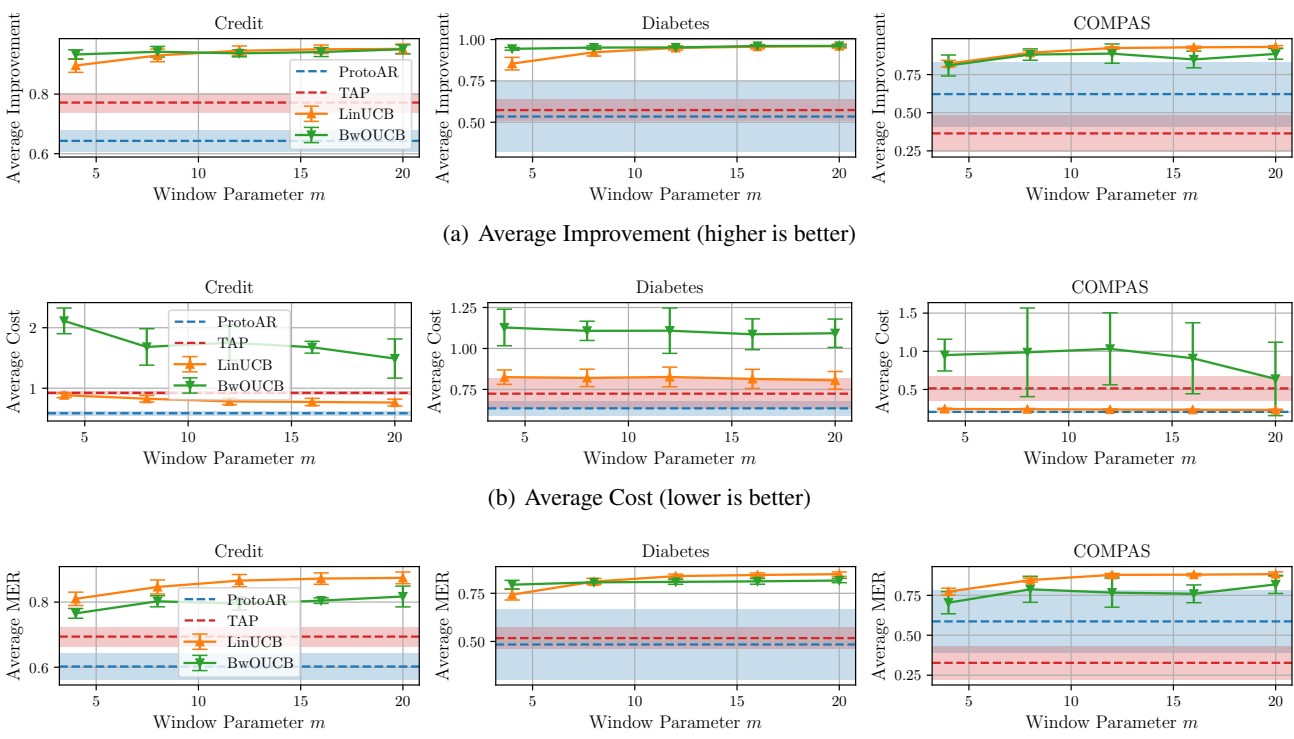

(a) Average Improvement (higher is better)

(b) Average Cost (lower is better)

(c) Average Mean Expected Reward of Final Round (higher is better)

*Figure 20.* Sensitivity analyses of the window parameter $m$ for NN.

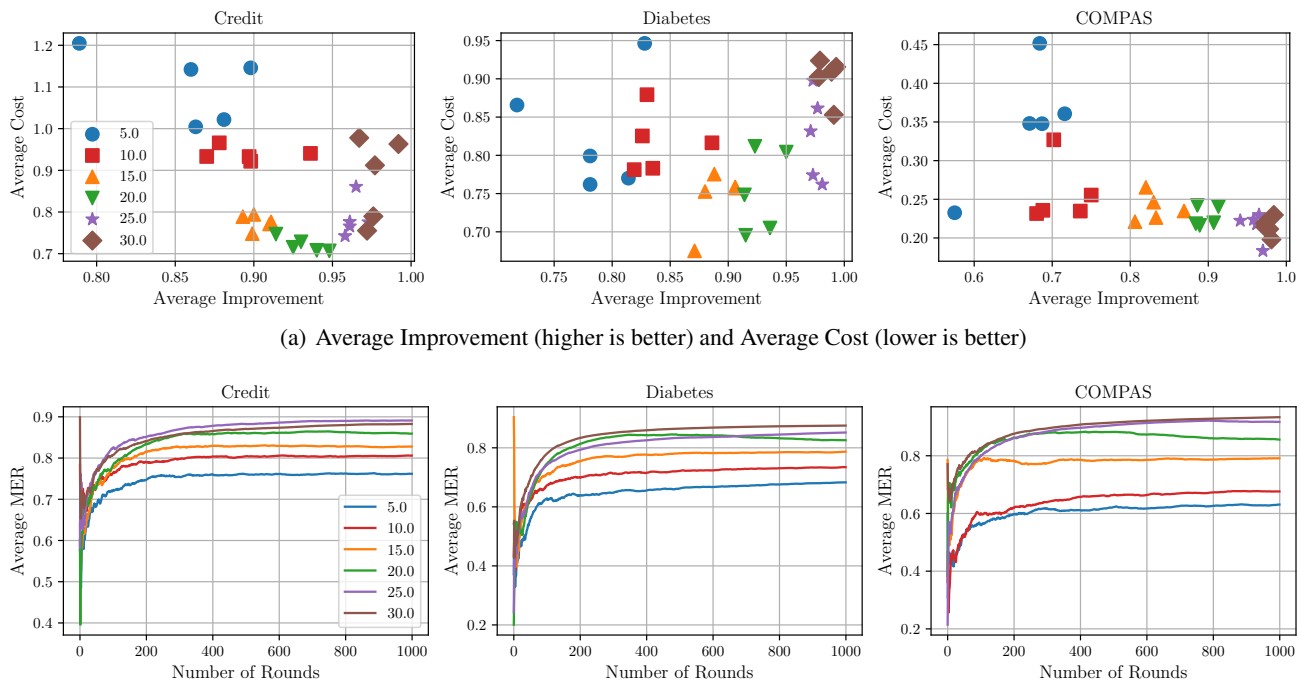

(a) Average Improvement (higher is better) and Average Cost (lower is better)

(b) Average Mean Expected Reward in Each Round (higher is better)

*Figure 21.* Sensitivity analyses of the regularization parameter $\lambda$ of Algorithm 1 for RF under the noiseless cost evaluation situation.

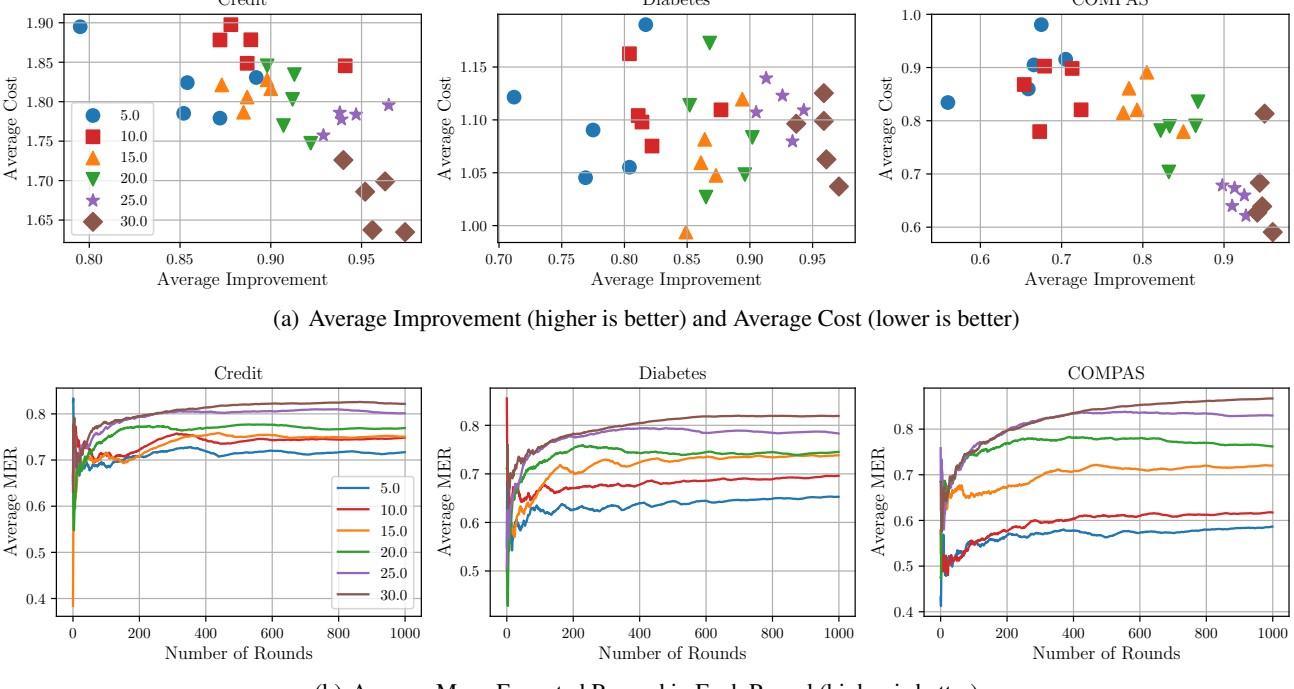

(a) Average Improvement (higher is better) and Average Cost (lower is better)

(b) Average Mean Expected Reward in Each Round (higher is better)

*Figure 22.* Sensitivity analyses of the regularization parameter $\lambda$ of Algorithm 1 for RF under the noisy cost evaluation situation with the noise level $\xi = 0.25$.

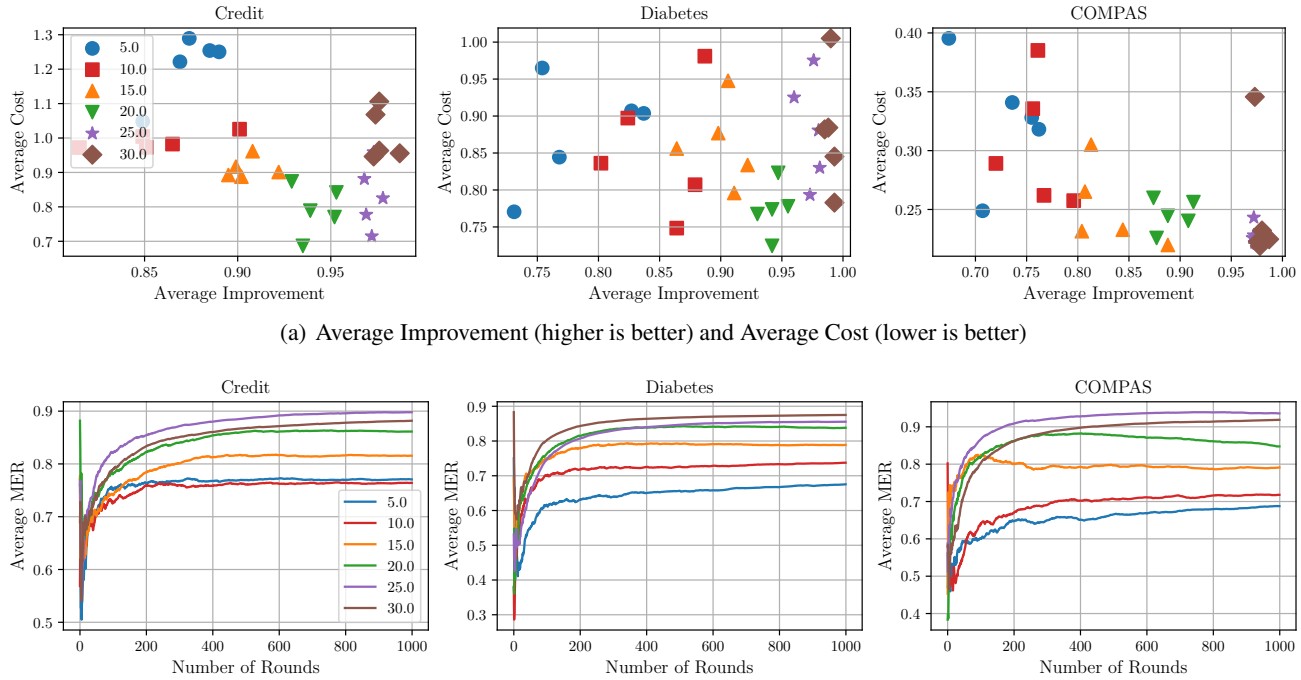

(a) Average Improvement (higher is better) and Average Cost (lower is better)

(b) Average Mean Expected Reward in Each Round (higher is better)

*Figure 23.* Sensitivity analyses of the regularization parameter $\lambda$ of Algorithm 1 for NN under the noiseless cost evaluation situation.

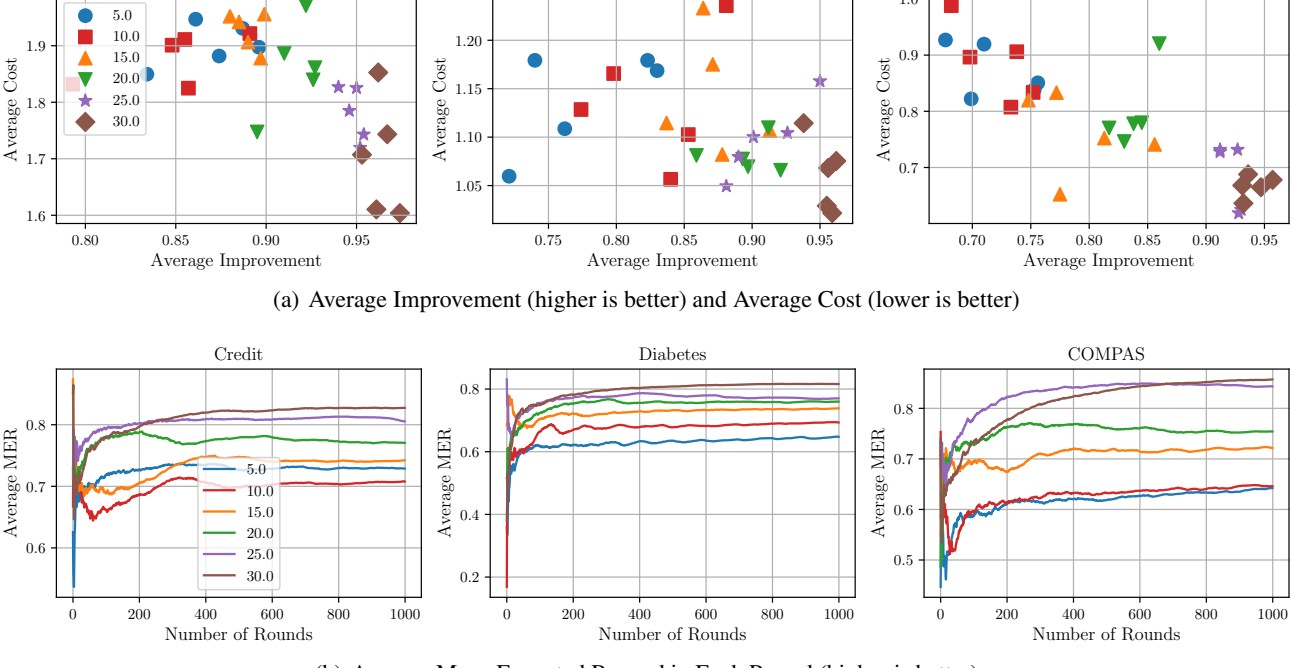

(a) Average Improvement (higher is better) and Average Cost (lower is better)

(b) Average Mean Expected Reward in Each Round (higher is better)

*Figure 24.* Sensitivity analyses of the regularization parameter $\lambda$ of Algorithm 1 for NN under the noisy cost evaluation situation with the noise level $\xi = 0.25$.

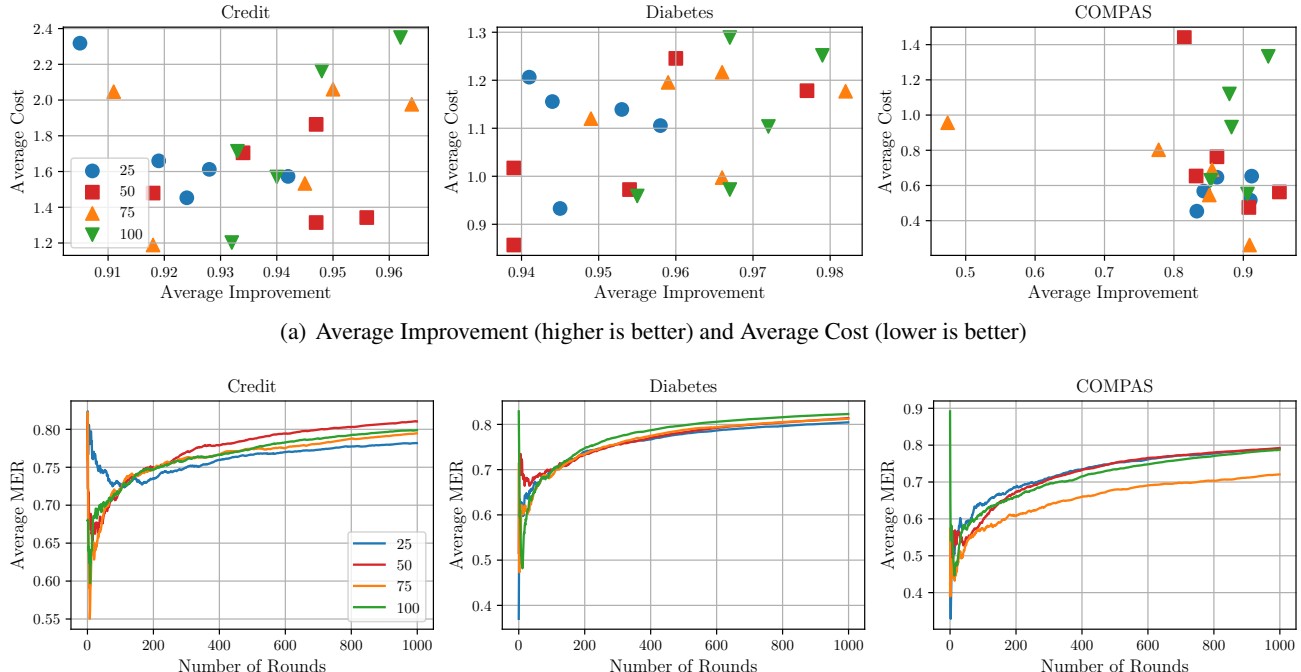

(a) Average Improvement (higher is better) and Average Cost (lower is better)

(b) Average Mean Expected Reward in Each Round (higher is better)

*Figure 25.* Sensitivity analyses of the total number of trees $B$ of Algorithm 2 for RF.

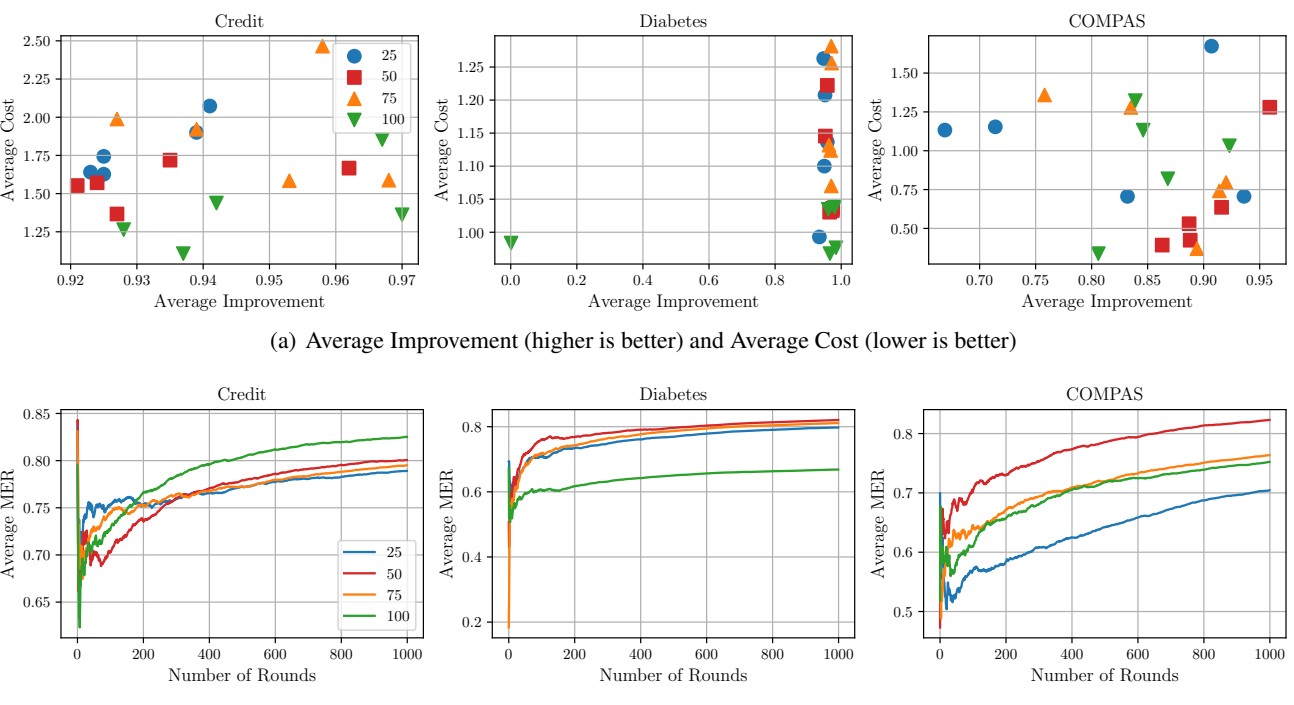

(a) Average Improvement (higher is better) and Average Cost (lower is better)

(b) Average Mean Expected Reward in Each Round (higher is better)

*Figure 26.* Sensitivity analyses of the total number of trees $B$ of Algorithm 2 for NN.

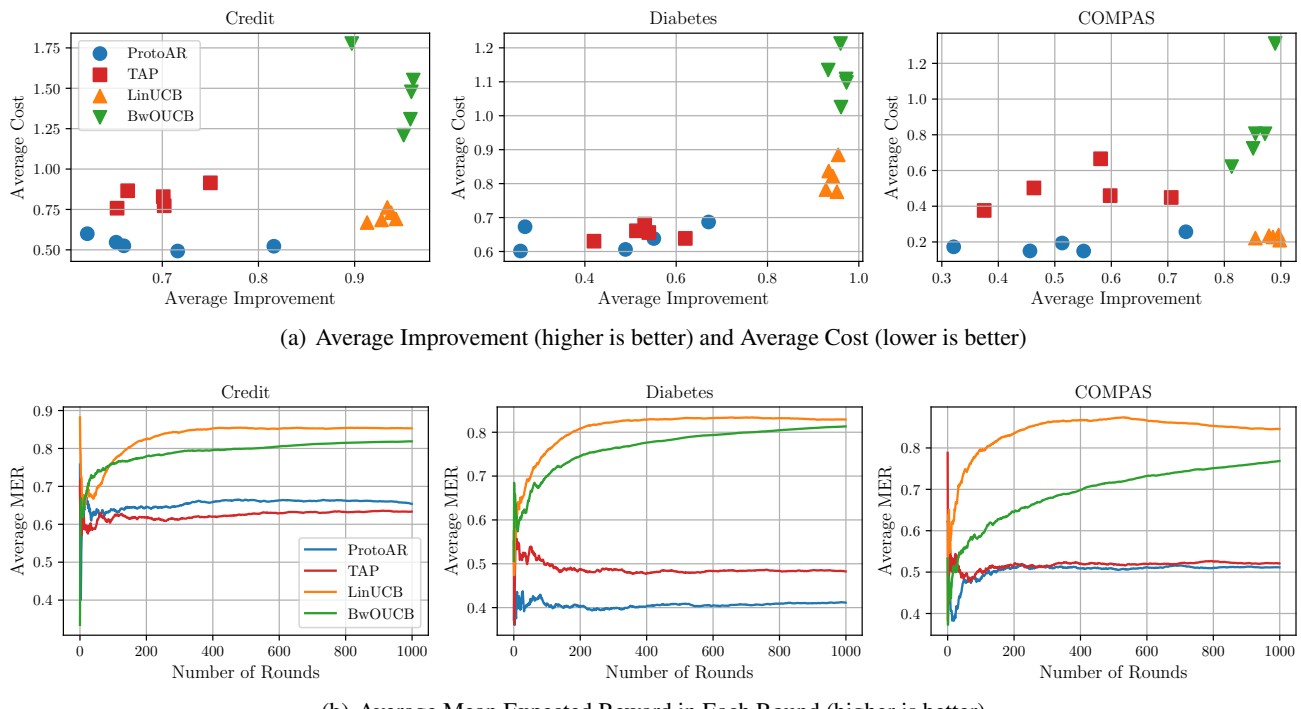

(a) Average Improvement (higher is better) and Average Cost (lower is better)

(b) Average Mean Expected Reward in Each Round (higher is better)

*Figure 27.* Experimental results of the model update scenario for RF where the update frequency is once every 25 rounds.

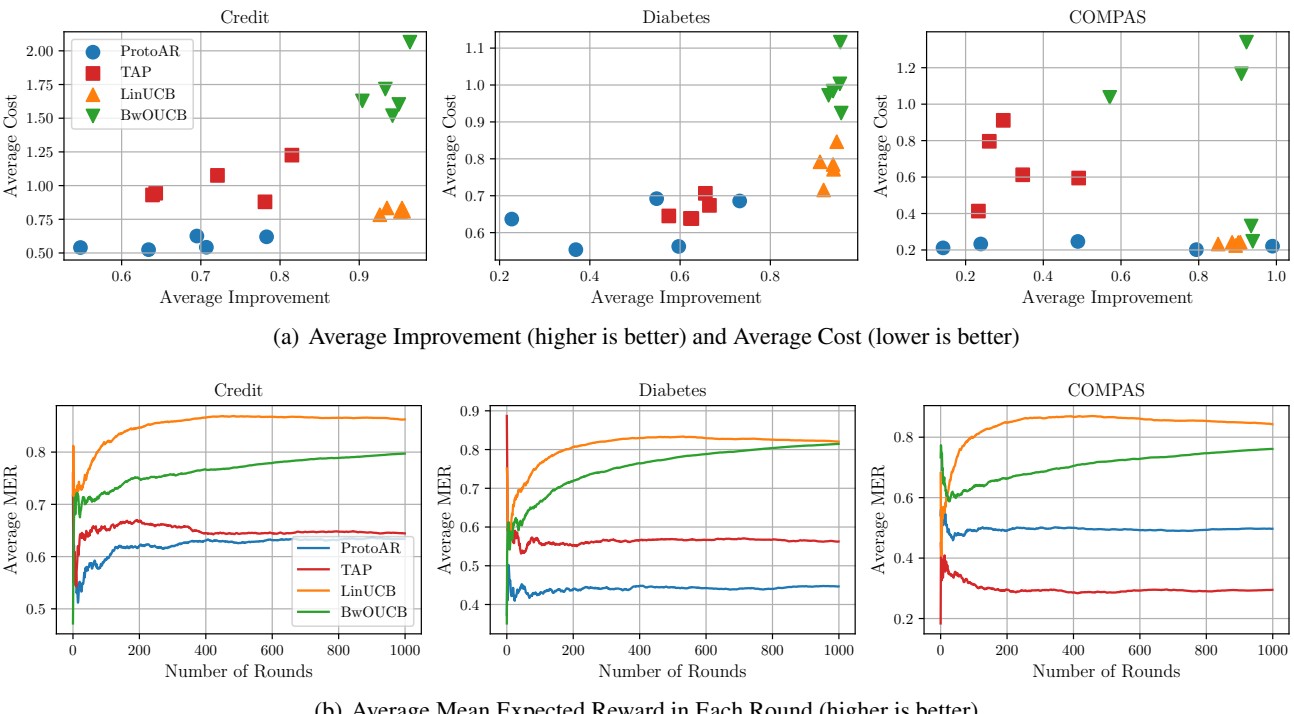

(a) Average Improvement (higher is better) and Average Cost (lower is better)

(b) Average Mean Expected Reward in Each Round (higher is better)

*Figure 28.* Experimental results of the model update scenario for NN where the update frequency is once every 25 rounds.

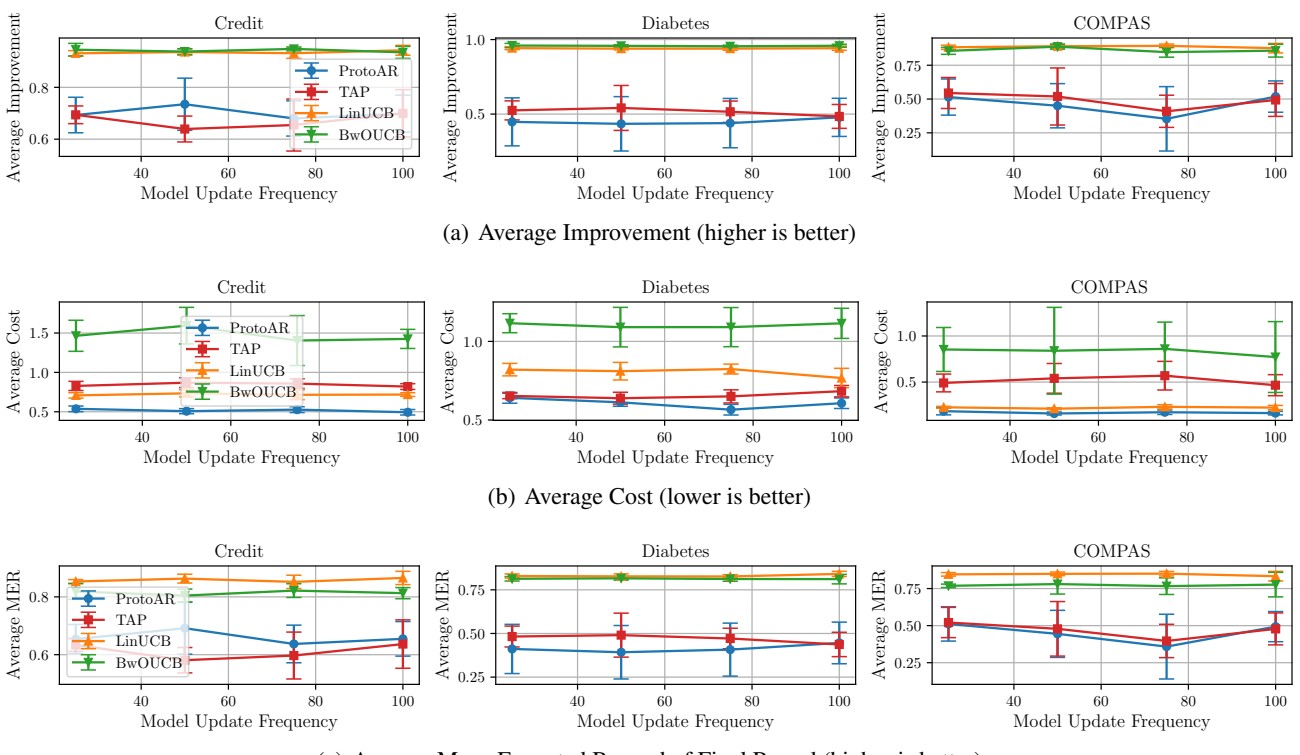

*Figure 29.* Sensitivity analyses of the model update frequency for RF.

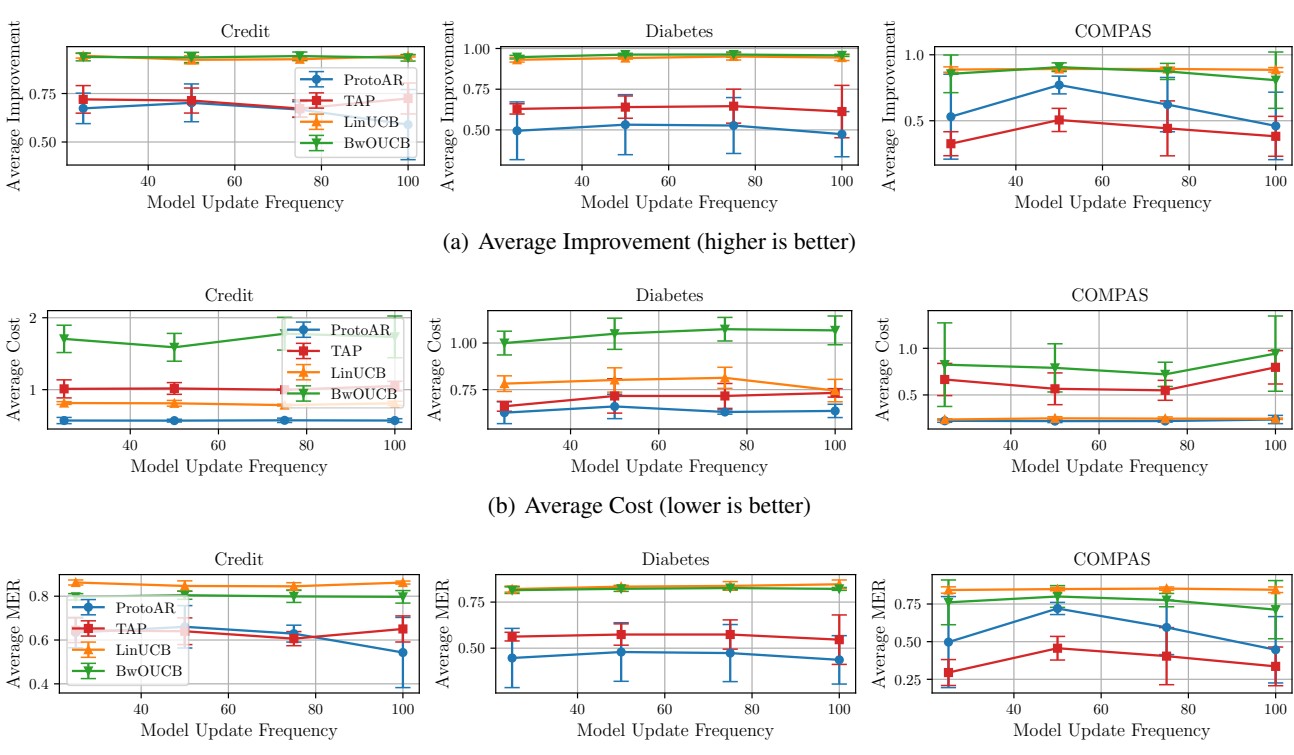

*Figure 30.* Sensitivity analyses of the model update frequency for NN.

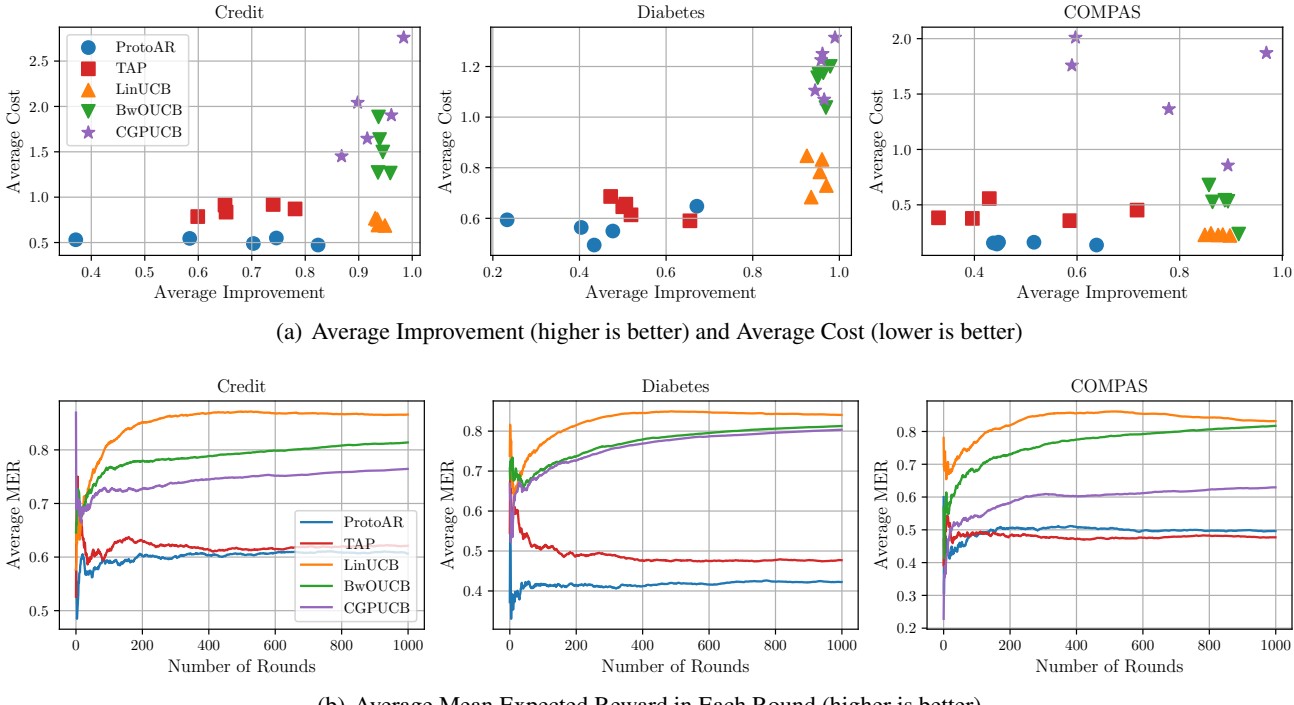

(a) Average Improvement (higher is better) and Average Cost (lower is better)

(b) Average Mean Expected Reward in Each Round (higher is better)

*Figure 31.* Experimental results of comparison to the algorithm based on the Gaussian process (CGPUCB) for RF.

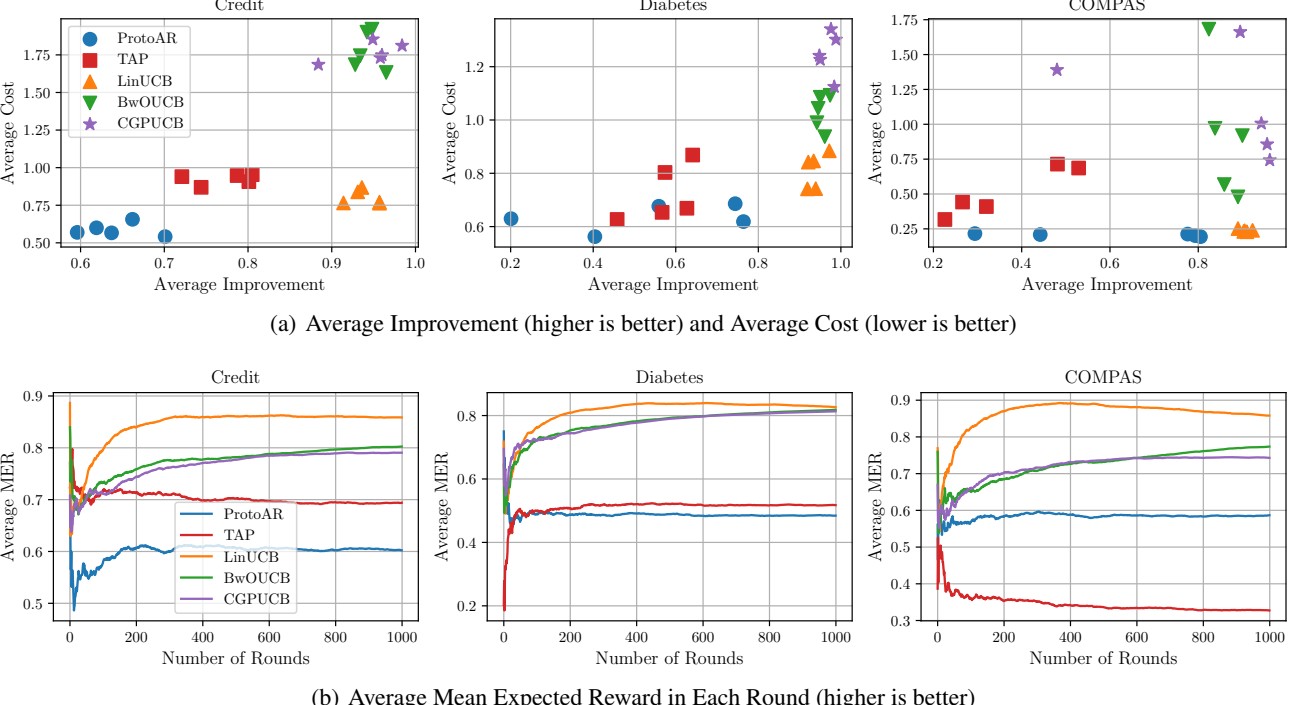

(a) Average Improvement (higher is better) and Average Cost (lower is better)

(b) Average Mean Expected Reward in Each Round (higher is better)

*Figure 32.* Experimental results of comparison to the algorithm based on the Gaussian process (CGPUCB) for NN.

*Table 2.* Experimental results on the average running time [s] per round of each method. We can see that the running time of LinUCB was not significantly different from those of ProtoAR and TAP. In addition, BwOUCB was significantly faster than CGPUCB, while their performance was close to each other.

(a) RF

| Dataset | ProtoAR | TAP | LinUCB | BwOUCB | CGPUCB |
|---|---|---|---|---|---|
| Credit | $0.000496 \pm 0.0002$ | $0.00045 \pm 0.0001$ | $0.00099 \pm 0.0023$ | $0.01782 \pm 0.0125$ | $4.524303 \pm 3.3757$ |
| Diabetes | $0.000406 \pm 0.0002$ | $0.000389 \pm 0.0$ | $0.000952 \pm 0.0046$ | $0.017913 \pm 0.0124$ | $4.984736 \pm 3.955$ |
| COMPAS | $0.000447 \pm 0.0006$ | $0.000379 \pm 0.0001$ | $0.000871 \pm 0.0028$ | $0.017919 \pm 0.0126$ | $4.92054 \pm 3.8675$ |

(b) NN

| Dataset | ProtoAR | TAP | LinUCB | BwOUCB | CGPUCB |
|---|---|---|---|---|---|
| Credit | $0.000506 \pm 0.0002$ | $0.00053 \pm 0.0002$ | $0.000936 \pm 0.0024$ | $0.017705 \pm 0.0128$ | $4.813339 \pm 3.7733$ |
| Diabetes | $0.000439 \pm 0.0003$ | $0.000443 \pm 0.0002$ | $0.000835 \pm 0.0026$ | $0.017724 \pm 0.0126$ | $4.765345 \pm 3.5786$ |
| COMPAS | $0.000427 \pm 0.0003$ | $0.00038 \pm 0.0003$ | $0.000938 \pm 0.0027$ | $0.017427 \pm 0.0124$ | $5.024779 \pm 3.896$ |

