# OpenReview forum: "Algorithmic Recourse for Long-Term Improvement"
_ICML.cc/2025/Conference — ICML 2025 poster_

### Official Review · Reviewer_3WCv · 2025-03-18

**Overall Recommendation:** 3

**Summary:**

Existing work in improvement-oriented algorithmic recourse assumes access to an accurate underlying model of whether or not a user taking an action improves their outcome. The paper proposes to overcome this limitation by using a bandit algorithm to learn a more accurate improvement model over time based on delayed feedback, while assuming that the action costs are known. The paper reduces this problem to a contextual bandit problem with delayed feedback, and proves that the resulting algorithm asymptotically achieves zero regret. They then propose a heuristic method to account for unknown costs. They then compare to class prototype-based algorithmic recourse, trustworthy actionable perturbation, and LinUCB.

**Claims And Evidence:**

1) The paper claims that the task can be reduced to a contextual linear bandit problem with delayed feedback. This is supported by Proposition 4.1.

2) The paper claims that LinUCB can solve the problem well when the costs are known. This is supported by the experiments in Figures 2, 3, and 4 though primarily 2.

3) The paper claims that a proposed heuristic method can solve the problem well, as supported by the empirical results.

**Essential References Not Discussed:**

I am unaware of any critical missing references, though I am not an expert in the algorithmic recourse literature.

**Experimental Designs Or Analyses:**

The experimental designs seemed valid, though the choice of baseline is a critical question that I'm not familiar enough with the literature to comment on further.

**Methods And Evaluation Criteria:**

Yes, though the choice of baseline is critical for making this argument and I am not familiar enough with the recourse literature to evaluate it.

**Other Comments Or Suggestions:**

n/a

**Other Strengths And Weaknesses:**

Originality: This paper, as far as I know, makes original contributions in combining algorithmic recourse with contextual bandits with delayed feedback.

Significance: The results are significant, improving over current algorithmic recourse methods.

Clarity: The paper is generally clear, though it could be clarified what is meant by "long term perspective".

**Questions For Authors:**

1) What specifically is meant by "long-term perspective"? The phrase comes up repeatedly, but it never gets defined explicitly, and seems to mean something like "the oracle used to estimate improvement becomes more correct over time". This question is pretty minor, but could help me better understand the significance of the results.
2) Isn't the LinUCB formulation somewhat misspecified? The outcome labels seem to be binary which is fine for LinUCB formally, but it seems like you could probably get better performance by using a more suited model.

**Relation To Broader Scientific Literature:**

They clearly show their relationship to contextual bandits with delayed feedback, as well as the algorithmic recourse literature.

**Theoretical Claims:**

I did not carefully check the

---

> ### Author Rebuttal · Authors · 2025-04-01
>
> We would like to thank the reviewer for valuable and thoughtful feedback. We will reflect all of them in our final version. In the following, we will respond to the key comments and questions raised by the reviewer.
>
> ---
>
> > What specifically is meant by "long-term perspective"? The phrase comes up repeatedly, but it never gets defined explicitly, and seems to mean something like "the oracle used to estimate improvement becomes more correct over time". This question is pretty minor, but could help me better understand the significance of the results.
>
> Thank you for your important question. As you pointed out, the phrase "long-term perspective" in this paper refers to the idea that our framework becomes to accurately estimate the improvement of recourse actions over time as we obtain more feedback. In our final version, we will clarify and emphasize this point to avoid any ambiguity.
>
>
> > Isn't the LinUCB formulation somewhat misspecified? The outcome labels seem to be binary which is fine for LinUCB formally, but it seems like you could probably get better performance by using a more suited model.
>
> Thank you for your insightful comment regarding our model. Because our algorithm is based on the OTFLinUCB algorithm (Vernade et al. 2020) that is tailored for the formulation where the outcome is binary, we think our LinUCB formulation is not misspecified. Of course, we acknowledge the potential to get better performance by designing more suited models and algorithms for our problem, which is an important direction for future research.
>
> ---
>
> We hope that we have adequately addressed all your questions and concerns. Please let us know if we can provide any further details and/or clarifications. Thank you again for your valuable feedback.

---

### Official Review · Reviewer_aPEN · 2025-03-18

**Overall Recommendation:** 3

**Summary:**

This paper proposes an online algorithmic recourse setting where an unknown oracle returns the real-world outcome for a given input. The authors introduce bandit algorithms to address this problem. Experiments demonstrate that their proposed methods outperform existing recourse approaches.

**Claims And Evidence:**

As far as I can tell, the claims made in the paper are supported by theoretical and empirical evidence.

**Essential References Not Discussed:**

Not to my knowledge.

**Experimental Designs Or Analyses:**

I think the experiment procedure is appropriate.

**Methods And Evaluation Criteria:**

The method is solid and makes sense, although it might lack novelty as this is a direct application from the bandit algorithms.

**Other Comments Or Suggestions:**

NA

**Other Strengths And Weaknesses:**

I find this paper addressing a novel algorithmic recourse problem, which is interesting. I also find the paper to be very well-written.

The main concern I have is the real-world application (i.e., why should we care about the long-term improvement?). The algorithmic recourse algorithm operates under the assumption that the underlying model $h$ describes the whole world (which is an unrealistic assumption rightly pointed out by the authors). In other words, the real $h^*$ does not really matter, because it is $h$ which makes the decision. Let's consider the loan application scenario, and let $x'$ be the generated recourse of $x$. I would argue that the bank should always grant the loan if $h(x')=1$, even if in reality, $h^*(x')=0$, and in such case, it might be more important to make $h \approx h^*$, rather than creating a new method to take into account this scenario.

In addition, I would also argue that such an online setting is definitely undesirable for the banks. If the user takes the first action $a_1$, and it results in $h^*(x+a_1)=0$ (meaning that the user defaults the loan), the bank will undertake the loss.

I would suggest the authors to come up with a concrete scenario that optimizing for the long-term improvements is useful and desirable.

**Questions For Authors:**

1. I am interested in learning how the recourse are generated for the categorical features, as they are one-hot encoded.

**Relation To Broader Scientific Literature:**

To my knowledge, this paper addressed a novel problem.

**Theoretical Claims:**

I glanced through the theoretical claims, and I did not catch an error.

---

> ### Author Rebuttal · Authors · 2025-04-01
>
> We would like to thank the reviewer for valuable and thoughtful feedback. We will reflect all of them in our final version. In the following, we will respond to the key comments and questions raised by the reviewer.
>
> ---
>
> > The main concern I have is the real-world application (i.e., why should we care about the long-term improvement?). The algorithmic recourse algorithm operates under the assumption that the underlying model $h$ describes the whole world (which is an unrealistic assumption rightly pointed out by the authors). In other words, the real $h^\ast$ does not really matter, because it is $h$ which makes the decision. Let's consider the loan application scenario, and let $x'$ be the generated recourse of $x$. I would argue that the bank should always grant the loan if $h(x') = 1$, even if in reality, $h^\ast(x') = 0$, and in such case, it might be more important to make $h \approx h^\ast$, rather than creating a new method to take into account this scenario.
>
> Thank you for your insightful comment. We agree with the reviewer's comment that banks should grant the loan for $x'$ with $h(x') = 1$, even if $h^\ast(x') = 0$, and that making $h$ close to $h^\ast$ might be a direct solution. However, we believe that filling the gap between $h$ and $h^\ast$ completely is unrealistic due to various factors such as data limitations, model complexity, and evolving real-world conditions. Therefore, we argue that it is still valuable to consider providing improvement-oriented actions even with the inherent limitations of $h$.
> In addition, we conducted experiments in Appendix B.5 where we iteratively updated $h$ using the obtained feedbacks $(x', h^\ast(x'))$ as new training samples. We observed that the existing baselines often failed to provide recourses $x'$ such that $h^\ast(x') = 1$ even with the updated $h$. This suggests that making $h$ sufficiently close to $h^\ast$ is still challenging in real-world scenarios.
> In summary, our method contributes to bridging the gap between the practical necessity of using $h$ and the ideal scenario of knowing $h^\ast$. We will add the above discussion to our final version.
>
>
> > In addition, I would also argue that such an online setting is definitely undesirable for the banks. If the user takes the first action $a_1$, and it results in $h^\ast(x + a_1) = 0$ (meaning that the user defaults the loan), the bank will undertake the loss.
> > I would suggest the authors to come up with a concrete scenario that optimizing for the long-term improvements is useful and desirable.
>
> Thank you for your important suggestion. As you mentioned, the banks may undertake the losses $h^\ast(x_t+a_t) \not= 1$ during the early phase $t$ where our algorithm explores improvement-oriented actions. However, we note that this issue with early losses is indeed the reason why we focus on minimizing the entire regret with respect to the loss over time $t = 1, 2, \dots, T$. By explicitly optimizing for long-term improvement, we aim to balance the exploration of potentially beneficial actions with the need to mitigate immediate losses. From this perspective, we think that the loan application scenario you raised is a concrete scenario where optimizing long-term improvements is useful and desirable. While there may be initial defaults, the long-term benefits of guiding users towards actions that genuinely improve their repayment ability will outweigh the short-term risks. We will emphasize this point in our final version.
>
>
> > I am interested in learning how the recourse are generated for the categorical features, as they are one-hot encoded.
>
> In our experiments, we employ the recourse generation algorithm based on the class prototypes (Van Looveren & Klaise, 2021). For an input instance $x$, this approach finds a recourse instance $\tilde{x}$ from a subset of a training set such that $h(\tilde{x}) = 1$ and that the input $x$ can reach with the minimum cost $c(a \mid x)$, where $a = \tilde{x} - x$. Because recourse instances $\tilde{x}$ are selected from a training set, they inherently satisfy the one-hot constraints for categorical features. Consequently, the obtained recourse action $a$ naturally preserves the structure of categorical features. Note that we evaluate the cost of one-hot encoded features in the same manner as numerical ones and impose constraints on immutable features to prevent them from being altered by actions (e.g., gender and race in the COMPAS dataset).
>
> ---
>
> We hope that we have adequately addressed all your questions and concerns. Please let us know if we can provide any further details and/or clarifications. Thank you again for your valuable feedback.

---

> > ### Comment · Reviewer_aPEN · 2025-04-01
> >
> > I appreciate the author's reply to my question. Given that my main concern about the motivation of the work is still valid, I will keep my score.

---

### Official Review · Reviewer_f6gN · 2025-03-22

**Overall Recommendation:** 1

**Summary:**

This paper is about algorithmic recourse: it aims to help individuals take actions to change unfavorable predictions made by machine learning models (like getting a loan rejection changed to approval). The issue that this paper tries to resolve is that many current methods only focus on changing the prediction itself, without ensuring that the action actually improves the individual's real-world outcome (e.g., ensuring that the person can actually repay the loan).
The authors propose a new framework to suggest actions that not only change the prediction but also improve the real-world outcome over the long term. They use tools from bandit problems with delayed feedback (it may take some time to observe whether the action worked in the real world or not) and suggest actions accordingly. The paper compares two approaches—contextual linear bandit (section 4) and contextual Bayesian optimization (section 5)—for selecting these actions based on past outcomes. Some further comparison between two approaches are presented in section 5.3. Experiments show that their methods lead to better long-term improvement than two baselines (ProtoAR and TAP).

**Claims And Evidence:**

What is good:
- The paper states clearly the assumptions in each result.
- The theoretical result in Proposition 4.2 comes with a proof. The proof is correct.
- The claim that the proposed methods outperform the baselines are supported by the numerical results with several setting (noisy costs, etc.)

**Essential References Not Discussed:**

No. The paper has cited most relevant papers that I am aware of.

**Experimental Designs Or Analyses:**

The experiment settings are reasonable to me.

**Methods And Evaluation Criteria:**

No. The critical weak point of this paper is that the bandit setting with delayed feedback is not a good tool to address the recourse problem.

For a recourse problem, the individual may take a long time to implement the actions: in loan applications (the Credit dataset), whether the person will repay the loan or not is an event that is realized in a few years' horizon. Similarly, in healthcare (the Diabetes dataset), the patient may take years to get on with a healthy lifestyles.

The paper does not discuss the temporal horizon of the problem. I can see there is a huge conflict between the real-world recourse problem (delayed feedbacks measured in years) with the online learning setting (the recourse is generated at a daily frequency, and the learning horizon could be a few months, the feedback is still short-term). Once again, I need to emphasize that we are focusing on consequential domains, and the actions has to be impactful (in Credit, the person may need to save more money/reduce spending in a serious manner to meet the criteria), and we need to avoid hacking/gaming (in Credit, the person can instantaneously borrow money to meet the criteria -- this is considered cheating).

**Other Comments Or Suggestions:**

- The function $\phi$ is used in Algorithm 1, but $\phi$ is defined in the proof of Proposition 4.1. I recommend defining phi in the main text

**Other Strengths And Weaknesses:**

It is unclear to me why the authors would like to include Section 4. The assumptions of Section 4 is really strong (knowing $\nu$). Moreover, the algorithm of Section 5 seems to be better than LinUCB anyways. I can think that the authors include Section 4 because it has some theoretical support, however, the theoretical results therein is nearly a "copy-and-paste" result from the literature. For that, having Section 4 in the paper is a distraction.

I recommend the authors to study the theoretical guarantees of the contextual Bayesian algorithm in Section 5.

**Questions For Authors:**

1. Could the authors provide the readers with several reasonable scenarios where online learning could be blended with *consequential* decision making with long delayed feedbacks? This requires specifying the distribution $\mathcal D$ (see end of Section 3.1) and validating that the support of $\mathcal D$ is appropriate with the horizon $T$ of the online learning framework.

2. Is it possible to provide any guarantees for the algorithms proposed in Section 5?

**Relation To Broader Scientific Literature:**

Unclear. The paper is mainly about formulating the algorithmic recourse problem into the bandit with delayed feedback setting, and then use existing results from bandit to solve the problem.

There does not seem to be any direct contribution to the broader scientific literature.

**Theoretical Claims:**

The proof of Proposition 4.2 is correct, but it follows largely from Vernade et al. (2020).

There is no theoretical guarantee for Section 5, which is another weak point of this paper.

---

> ### Author Rebuttal · Authors · 2025-04-01
>
> We would like to thank the reviewer for valuable and thoughtful feedback. We will reflect all of them in our final version. In the following, we will respond to the key comments and questions raised by the reviewer.
>
> ---
>
> > **Methods And Evaluation Criteria**
> > No. The critical weak point of this paper is that the bandit setting with delayed feedback is not a good tool to address the recourse problem. [...] and we need to avoid hacking/gaming [...].
>
> Thank you for your important comment. We acknowledge the inherent challenges posed by the long-term nature of recourse implementation. However, we maintain that our bandit-based approach offers valuable contributions and can be a foundational step for the following reasons:
> - We are the first to demonstrate the feasibility of achieving long-term improvement by exploiting feedback in the recourse problem, even if only delayed feedback is available. This addresses real-world recourse challenges even when feedback is not given immediately.
> - While we recognize that effectively handling long delays remains a challenge, long delays do not preclude the potential of our method. Even if it takes a long time to observe the first feedback, our framework can work well once we start to observe feedback.
> - Since our method considers the real-world outcome $h^\ast(x + a)$, it avoids suggesting hacking/gaming actions that only meet the decision criteria of $h$.
>
> > **Questions For Authors**  1. Could the authors provide the readers with several reasonable scenarios [...] validating that the support of $\mathcal{D}$ is appropriate with the horizon $T$ of the online learning framework.
>
> We summarize three scenarios where an online learning approach could be suitable for consequential decision-making:
>
> | Task | User / Decision Maker | Outcome | Action | Delay |
> | - | - | - | - | - |
> | Healthcare | Patient / Doctor | Physiological indicator (blood pressure or blood sugar level) | Dietary restrictions, exercise routine | 1--4 weeks |
> | Job Hiring | Job Seeker / Staffing Agency | Whether a job seeker is employed by a company to which he or she has applied | Regime revision, interview preparation | 4--6 weeks |
> | Human Resource | Employee / Company | Employee attrition within months | Reducing overtime, getting counselling |  3--6 months |
>
> In each case, round $t$ corresponds to a decision for a user $x_t$, and $T$ is the total number of users who receive actions, not elapsed real time. Let us consider the Healthcare task, for example. If the doctor provides actions with five patients every weekday and we deploy our method for at least one year, the horizon $T$ exceeds $1000$. Meanwhile, the maximum delay $D_t \sim \mathcal{D}$ is at most about $100$ (corresponding to four weeks), which is small compared to the overall horizon $T$. In such scenarios, we believe the delay distribution $\mathcal D$ remains appropriate throughout $T$, and our method can effectively provide improvement-oriented actions within a realistic time frame by collecting feedback from past users.
>
> > **Other Strengths And Weaknesses**
> > It is unclear to me why the authors would like to include Section 4. [...] For that, having Section 4 in the paper is a distraction.
>
> Thank you for your important comment. While we understand the reviewer's concern regarding Section 4, we still believe this section is essential for the following two reasons:
> - While we admit Proposition 4.2 is a direct application of the existing result by Vernade et al. 2020, we believe this proposition is valuable in clarifying the required assumptions to reduce Problem 1 to a mathematical model that can be solved with a theoretical guarantee.
> - We agree that knowing $\nu$ looks a strong assumption. However, even in the noisy cost situation of Section 6.3, which can be also interpreted as a misspecification of $\nu$, our algorithm of Section 4 outperformed existing baselines except for cost in Figure 3. Moreover, in early rounds until $t = 200$, it outperformed the algorithm of Section 5 in terms of MER in Figure 3(b). This suggests that our algorithm of Section 4 is practically robust to some degree of misspecification.
>
> > **Questions For Authors**
> > 2. Is it possible to provide any guarantees for the algorithms proposed in Section 5?
>
> Thank you for your important suggestion. Theoretical guarantees for Algorithm 2 are a valuable research direction. Replacing the BwO forest-based surrogate model with the Gaussian process, and leveraging the regret bound from Verma et al. (2022), might be possible. However, since their analysis relies on continuous outcomes, extending their results to our binary outcome setting is not trivial.
>
> ---
>
> We hope that we have adequately addressed all your questions and concerns. Please let us know if we can provide any further details and/or clarifications. Thank you again for your valuable feedback.

---

### Official Review · Reviewer_NJwZ · 2025-03-24

**Overall Recommendation:** 4

**Summary:**

The paper studies Algorithmic Recourse (AR), i.e., providing a recourse action $a$ to individuals $x$ to improve so that their classification changes from $h(x)=0$ to $h(x+a)=1$ or “h-valid.”  The authors frame their problem in the “improvement” setting (König et al.), where the goal is also to improve classification on some unknown classification oracle $h^*$ that correctly captures long-term improvement; thus, they want $h^*(x+a) = 1$ too.

They study this problem in the online setting, where in round $t$, the agent i) gets an individual $x_t$, ii) selects an action from the feasible actions $A_t$ (based on $x_t$ manipulation cost and whether they are h-valid), and iii) gets a reward related to being $h^*$-valid. This reward is delayed and revealed after some rounds (the delay distribution is unknown).

The authors propose two algorithms: contextual linear bandits (CLB) and contextual Bayesian optimization (CBO), which are used to solve the problem when the costs are known/unknown, respectively, and provide a regret bound for the CLB approach. They then run experiments on three datasets and compare these two methods against two baselines, showing improvement in reward and lower average cost.

**Claims And Evidence:**

Yes.

**Essential References Not Discussed:**

N/A

**Experimental Designs Or Analyses:**

Yes, the experiments are sound and the supplementary code has sufficient details.

**Methods And Evaluation Criteria:**

Yes the 3 datasets seem to be standard for algorithmic recourse problems. The experimental description and details are thorough.

**Other Comments Or Suggestions:**

N/A

**Other Strengths And Weaknesses:**

Strengths

1. The paper is well written, and the experimental evaluation is thorough.

2. The most insightful sections are the reduction to CLB in Section 4 and the BwO-based optimization in Section 5.1 to speed up CBO.

Weaknesses

1. The Bernoulli reward model in equation (2) needs more description. Can you mention what exactly in Fokkema et al. you’re referencing for $E(a|x)$? This seems to be a softmax sample of actions inversely proportional to cost. Also, can the reward be relaxed to something not distributional, e.g., simply the output of $h^*(x_t + a_t)$, which is revealed in a later round?

2. On a similar note, can you expand on the motivations behind the $P(Y=1| X=x_n)$ in the experiment Protocol paragraph, bullet point 3? How is this evaluated on the test instances from bullet point 4? Do you assume the test instances can shift only to one of the recourse instances?

**Questions For Authors:**

What is the impact of the stochastic delay distribution on your results, i.e., can you point out where it’s appearing in Proposition 4.2? Also, what can you say about fixed deterministic delays modeling some practical cases, e.g., when credit defaults are evaluated every month?

**Relation To Broader Scientific Literature:**

The paper is a good contribution to AR, and the online improvement setting is novel.

**Theoretical Claims:**

Yes, the claims for CLB are sound and have a proof of Proposition 4.2 in the Appendix.

---

> ### Author Rebuttal · Authors · 2025-04-01
>
> We would like to thank the reviewer for valuable and thoughtful feedback. We will reflect all of them in our final version. In the following, we will respond to the key comments and questions raised by the reviewer.
>
> ---
>
> > The Bernoulli reward model in equation (2) needs more description. Can you mention what exactly in Fokkema et al. you’re referencing for $E(a \mid x)$? This seems to be a softmax sample of actions inversely proportional to cost. Also, can the reward be relaxed to something not distributional, e.g., simply the output of $h^\ast(x_t + a_t)$, which is revealed in a later round?
>
> Thank you for your important comment on our reward model.
> We define $E(a \mid x)$ as the probability that an instance $x$ executes an action $a$ and assume that it decreases depending on the cost $c(a \mid x)$, as you pointed out. In particular, we define $E(a \mid x) = \exp (-\nu \cdot c(a \mid x))$ so that it decreases exponentially in $c(a \mid x)$. This definition is also employed by Fokkema et al. to model the probability that $x$ executes $a$.
> In addition, our reward model $R_t$ can be relaxed to a model that simply outputs $h^\ast(x_t + a_t)$. It corresponds to a special case of our Bernoulli reward model where we set $E(a_t \mid x_t) = 1$ and $I(a_t \mid x_t) = h^\ast(x_t + a_t)$ for any $x_t$ and $a_t$.
>
>
> > On a similar note, can you expand on the motivations behind the $P(Y=1 \mid X=x_n)$ in the experiment Protocol paragraph, bullet point 3? How is this evaluated on the test instances from bullet point 4? Do you assume the test instances can shift only to one of the recourse instances?
>
> Thank you for your important comment on our experimental protocol. As you said, for each test instance $x_t$, we restricted its candidate actions $a_t$ to those that can shift to one of the recourse instances $\tilde{\mathcal{X}}$. If we allow arbitrary actions $a_t$ for $x_t$, we can not evaluate the improvement of $a_t$ for $x_t$ because we do not know the oracle outcome $h^\ast(x_t + a_t)$ in real datasets. Our motivation to restrict candidates to the actions leading to recourse instances is to ensure the existence of a recourse instance $x_n$ such that $x_n = x_t + a_t$. This allows us to evaluate $h^\ast(x_t + a_t)$ using the label $y_n$ associated with $x_n$ as a proxy for the oracle outcome. In addition, to simulate a noisy oracle, we set the probability of improvement $P(Y=1 \mid X=x_t+a_t)$ using the value of $y_n$ with a noise $\varepsilon \sim \mathcal{N}(0, 1)$ and scaling the result to the range $[0, 1]$. In our final version, we will clarify this point and provide more description.
>
>
> > What is the impact of the stochastic delay distribution on your results, i.e., can you point out where it’s appearing in Proposition 4.2? Also, what can you say about fixed deterministic delays modeling some practical cases, e.g., when credit defaults are evaluated every month?
>
> Thank you for your important question.
> The delay distribution $\mathcal{D}$ impacts the term $\tau_m = P(D_1 \leq m)$ in our bound. This term increases as the delay $D_1 \sim \mathcal{D}$ for the first instance $x_1$ tends to be smaller than the window parameter $m$ of our algorithm, which makes our bound in Proposition 4.2 better. In essence, the more quickly feedback is received, the better our algorithm performs.
> In addition, if the delay is a fixed value and we know it in advance, we can set our window parameter $m$ to the value. This adaptation would likely lead to improved performance compared to the stochastic delay setting.
>
> ---
>
> We hope that we have adequately addressed all your questions and concerns. Please let us know if we can provide any further details and/or clarifications. Thank you again for your valuable feedback.

---

### Decision · Program_Chairs · 2025-05-01

**Decision:**

Accept (poster)

**Comment:**

The paper studies algorithmic recourse in a setting where the goal is to both be approved by the classifier and achieve the desired true label. They model this problem as a delayed feedback and use tools from bandit literature to provide no regret algorithms.

This is an important problem and studies an overlooked aspect of the original recourse problem. The reviewers generally liked the paper and its contribution. They found the paper well written and the experimental setting to be convincing.

The reviewers also had some concerns about the reasonability of the assumption (for example, the long feedback delay) or straightforward applications of previously studied bandit settings. I encourage the authors to discuss all of these concerns in the limitation section.